# RIGHT ON TIME: REVISING TIME SERIES MODELS BY CONSTRAINING THEIR EXPLANATIONS

## ABSTRACT

The reliability of deep time series models is often compromised by their tendency to rely on confounding factors, which may lead to incorrect outputs. Our newly recorded, naturally confounded dataset named P2S from a real mechanical production line emphasizes this. To avoid "Clever-Hans" moments in time series, i.e., to mitigate confounders, we introduce the method Right on Time (RioT). RioT enables, for the first time interactions with model explanations across both the *time* and *frequency* domain. Feedback on explanations in both domains is then used to constrain the model, steering it away from the annotated confounding factors. The dual-domain interaction strategy is crucial for effectively addressing confounders in time series datasets. We empirically demonstrate that RioT can effectively guide models away from the wrong reasons in P2S as well as popular time series classification and forecasting datasets.

## 1 INTRODUCTION

Time series data is ubiquitous in our world today. Everything that is measured over time generates some form of time series, for example, energy load (Koprinska et al., 2018), sensor measurements in industrial machinery (Mehdiyev et al., 2017) or recordings of traffic data (Ma et al., 2022). Various neural models are often applied to complex time series data (Ruiz et al., 2021; Benidis et al., 2023). As in other domains, these can be subject to confounding factors ranging from simple noise or artifacts to complex shortcut confounders (Lapuschkin et al., 2019). Intuitively, a confounder, also called "Clever-Hans" moment, can be a pattern in the data that is not relevant to the task but correlates with it during model training. A model can incorrectly pick up on this confounder and use it instead of the relevant features to e.g. make a classification. A confounded model does not generalize well to data without the confounder, which is a problem when employing models in practice (Geirhos et al., 2020). For time series, confounders and their mitigation have yet to receive attention, where existing works make specific assumptions about settings and data (Bica et al., 2020).

In particular, it is essential to mitigate shortcut confounders, i.e., spurious patterns in the training data used for the prediction. If a model utilizes confounding factors in the training set, its decision relies on wrong reasons and fails to generalize to unconfounded data. Model explanations play a crucial role in uncovering confounding factors, but they are not enough on their own to address them (cf. Fig. 1 **I**). While an explanation can reveal that the model relies on incorrect factors, it does not alter the model's outcome. To change this, we introduce Right on Time (RioT), a new method following the core ideas of the explanatory interactive learning (XIL) paradigm (Teso & Kersting, 2019), i.e., using feedback on explanations to mitigate confounders (cf. Fig. 1 **II**). RioT uses traditional explanation methods like Integrated Gradients (IG) (Sundararajan et al., 2017) to detect whether models focus on the right or the wrong time steps and utilizes feedback on the latter to revise the model.

However, confounding factors in time series data extend beyond the time domain. For example, a consistent noise frequency in an audio signal can act as a confounder without being tied to a specific point in time. RioT can handle these types of confounders by incorporating feedback in the frequency domain. To highlight the importance of mitigating confounders in time series data, we introduce a new real-world, confounded dataset called PRODUCTION PRESS SENSOR DATA (P2S). The dataset includes sensor measurements from an industrial high-speed press, essential to many manufacturing processes in the sheet metal working industry. The sensor data for detecting faulty production is naturally confounded and thus causes incorrect predictions after training. Next to its immediate

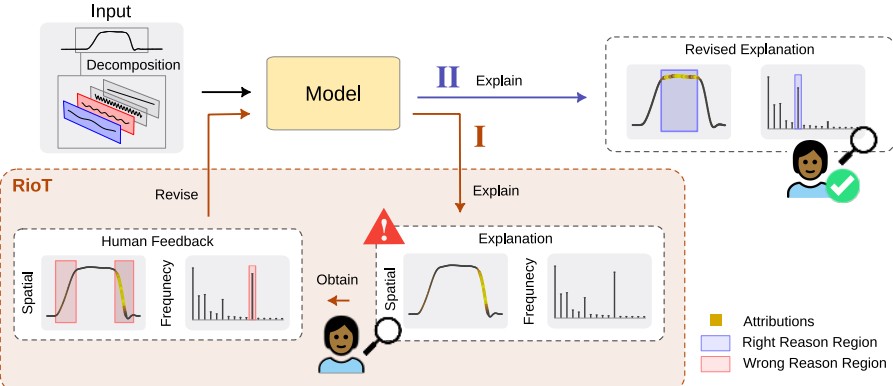

Figure 1: **I:** Explanations can reveal whether models rely on confounding factors in the input instead of relevant features. With RioT, a human can provide feedback on the spatial and frequency domain explanations for wrong reasons. This feedback is used to revise the model to not consider those regions. **II:** After revising via RioT, the model focuses on the right reasons instead.

industrial relevance, P2S is the first time series dataset that contains explicitly annotated confounders, enabling the evaluation and comparison of confounder mitigation strategies on real data.

Altogether, we make the following contributions: (1) We show both on our newly introduced real-world dataset P2S and on several other manually confounded datasets that SOTA neural networks on time series classification and forecasting can be affected by confounders. (2) We introduce RioT to mitigate confounders for time series data. The method can incorporate feedback not only on the time domain but also on the frequency domain. (3) By incorporating explanations and feedback in the frequency domain, we enable a new perspective on XIL, overcoming the important limitation that confounders must be spatially separable.[1]

The remainder of the paper is structured as follows. In Sec. 2, a brief overview of related work on explanations for time series and revising model mistakes is given. In Sec. 3, we introduce our method before providing an in-depth evaluation and discussion of the results in Sec. 4. Lastly, in Sec. 5, we conclude the paper and provide some potential avenues for future research.

## 2 RELATED WORK

**Explanations for Time Series.** Within the field of explainable artificial intelligence (XAI), various techniques to explain machine learning models and their outcomes have been proposed. While many techniques originate from image or text data, they were quickly adapted to time series Rojat et al. (2021). While backpropagation- and perturbation-based attribution methods provide explanations directly in the input space, other techniques like symbol aggregations (Lin et al., 2003) or shapelets (Ye & Keogh, 2011) aim to provide higher-level explanations. For a more in-depth discussion of explanations for time series, we refer to surveys by Rojat et al. (2021) or Schlegel et al. (2019). While it is essential to have explanation methods to detect confounding factors, they alone are insufficient to revise a model. Thus, explanations are the starting point of our method, as they enable users to detect confounders and provide feedback to overcome confounders in the model. In particular, we build upon Integrated Gradients (IG) (Sundararajan et al., 2017). This method computes attributions for the input by utilizing model gradients. We selected it because of its several desirable properties like completeness or implementation invariance and its wide use, also for time series data (Mercier et al., 2022; Veerappa et al., 2022).

**Explanatory Interactive Learning (XIL).** There is a growing field of research investigating confounding factors and how to overcome them. However, these mainly focus on visual data domains. One paradigm to overcome these confounders is using explanatory interactive learning, which describes the general process of revising a model's decision process based on human feedback(Teso & Kersting, 2019; Schramowski et al., 2020). A key element of XIL is using the model's explanations

---

[1]Code available at: https://anonymous.4open.science/r/RioT

to incorporate human feedback, thus revising the model's mistakes (Friedrich et al., 2023a). This is primarily done to revise models that show Clever-Hans-like behavior (being affected by shortcuts in the data) to prevent them from using these shortcuts Stammer et al. (2020). Several methods apply the XIL paradigm to image data. For example, Right for the Right Reasons (RRR) (Ross et al., 2017) and Right for Better Reasons (RBR) (Shao et al., 2021) use human feedback as a penalty mask on model explanations. Instead of penalizing wrong reasons, HINT (Selvaraju et al., 2019) rewards the model for focusing on the correct part of the input. Furthermore, Friedrich et al. (2023b) investigate the use of multiple explanation methods simultaneously. Although these methods are often employed to address confounders in image data, their application to time series data remains unexplored. To bridge this gap, we introduce RioT, a method that incorporates the core principles of XIL to the unique characteristics of time series data.

**Unconfounding Time Series.** Next to approaches from interactive learning, there is also some other work on unconfounding time series models. This line of work is generally based on causal analysis of the time series model and data (Flanders et al., 2011). Methods like Time Series Deconfounder (Bica et al., 2020), SqeDec (Hatt & Feuerriegel, 2024) or LipCDE (Cao et al., 2023), perform estimations on the data while mitigating the effect of confounders in covariates of the target variable. They generally mitigate the effect of the confounders through causal analysis and specific assumptions about the data generation. On the other hand, in this work we tackle confounders within the target variate, and have no further assumption besides that the confounder is visible in the explanations of the model, where these previous methods cannot easily be applied.

## 3 RIGHT ON TIME (RIOT)

The core intuition of Right on Time (RioT) is to leverage human feedback to steer a model away from wrong reasons. In that, the method follows the general paradigm of XIL. To make potential combinations with other methods within XIL straightforward, we develop RioT along the steps found by Friedrich et al. (2023a), namely *Select*, *Explain*, *Obtain* and *Revise*. In *Select*, instances for feedback and following model revision are selected. Following previous methods, we select all samples by default but also explore using only subsets of the data. Afterwards, *Explain* covers model explanations, before in *Obtain*, a human provides feedback on the selected instances. Lastly, in *Revise*, the feedback is integrated into the model to overcome the confounders. We introduce RioT along these steps in the following (as illustrated in Fig. 2). But let us first establish some notation for the remainder of this paper.

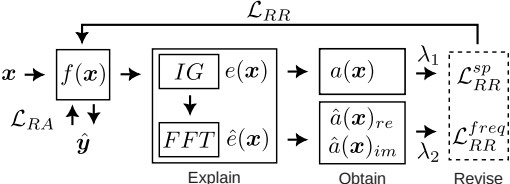

Figure 2: This figure depicts the flow of explanation and revision, beginning with input data $x$, through the model $f(x)$ to explanations $e(x)$, annotated feedback $a(x)$, and finally back to the model. IG provides the spatial model explanation, and is transformed via FFT to obtain the frequency explanation. Expert user annotations can be applied in either or both domains. They are utilized by the right-reason loss ($\mathcal{L}_{RR}^{sp}$ and $\mathcal{L}_{RR}^{freq}$) to guide the model away from confounders in both the time and frequency domains.

Given a dataset $(\mathcal{X}, \mathcal{Y})$ and a model $f(\cdot)$ for time series classification or forecasting. The dataset consists of $D$ many pairs of $x$ and $y$. Thereby, $x \in \mathcal{X}$ is a time series of length $T$, i.e., $x \in \mathbb{R}^T$. For $K$ class classification, the ground-truth output is $y \in \{1, \ldots, K\}$ and for forecasting, the ground-truth output is the forecasting window $y \in \mathbb{R}^W$ of length $W$. The ground-truth output of the full dataset is then described as $\mathcal{Y}$ in both cases. For a datapoint $x$, the model generates the output $\hat{y} = f(x)$, where the dimensions of $\hat{y}$ are the same as of $y$ for both tasks.

### 3.1 EXPLAIN

Given a pair of input $x$ and model output $\hat{y}$ for time series classification, the explainer generates an explanation $e_f(x) \in \mathbb{R}^T$ in the form of attributions to explain $\hat{y}$ w.r.t. $x$. For an element of the input, a large attribution value means a large influence on the output. In the remainder of the paper, explanations refer to the model $f$, but we drop $f$ from the notation to declutter it, resulting in $e(x)$. We use IG (Sundararajan et al., 2017) (Eq. 1) as an explainer, an established gradient-based attribution method. This method integrates the gradient along the path (using the integration variable $\alpha$) from

a baseline $\bar{x}$ to the input $x$ and multiplies the result with the difference between baseline and input. However, we make some adjustments to the base method to make it more suitable for time series and model revision, namely taking the absolute value of the difference between $\boldsymbol{x}$ and $\bar{\boldsymbol{x}}$ ( further details in App. A.1). In the following, we introduce the modifications to use attributions for forecasting and to obtain explanations in the frequency domain.

$$e(\boldsymbol{x}) = |\boldsymbol{x} - \bar{\boldsymbol{x}}| \cdot \int_0^1 \frac{\partial f(\tilde{\boldsymbol{x}})}{\partial \tilde{\boldsymbol{x}}}\Bigg|_{\tilde{\boldsymbol{x}} = \bar{\boldsymbol{x}} + \alpha(\boldsymbol{x} - \bar{\boldsymbol{x}})} d\alpha \quad (1) \qquad e(\boldsymbol{x}) = \frac{1}{W}\sum_{i=1}^{W} e_i'(\boldsymbol{x}) \quad (2)$$

**Attributions for Forecasting.** In a classification setting, attributions are generated by propagating gradients back from the model output (of its highest activated class) to the model inputs. However, there is often no single model output in time series forecasting. Instead, the model generates one output for each timestep of the forecasting window simultaneously. Naively, one could use these $W$ outputs and generate as many explanations $e_1'(\boldsymbol{x}), \ldots e_W'(\boldsymbol{x})$, where each $e_i'(\boldsymbol{x})$ is the IG explanation using the i-th time step from the forecasting window as target instead of a classification label. This number of explanations would, however, make it even harder for humans to interpret the results, as the size of the explanation increases with $W$(Miller, 2019). Therefore, we propose to aggregate the individual explanations by averaging in Eq. 2. Averaging attributions over the forecasting window provides a simple yet robust aggregation of the explanations. Other means of combining them, potentially even weighted based on distance of the forecast in the future are also imaginable. Overall, this allows attributions for time series classification and forecasting to be generated similarly.

**Attributions in the Frequency Domain.** Time series data is often given in the frequency representation. Sometimes, this format is more intuitive for humans to understand than the spatial representations. As a result, providing explanations in this domain is essential. Vielhaben et al. (2023) showed how to obtain frequency attributions of the method Layerwise Relevance Propagation (Bach et al., 2015), even if the model does not operate directly on the frequency domain. We transfer this idea to IG: for an input sample $\boldsymbol{x}$, we generate attributions with IG, resulting in $e(\boldsymbol{x}) \in \mathbb{R}^T$ (Eq. 1 for classification or Eq. 2 for forecasting). We then interpret the explanation as a time series, with the attribution scores as values. To obtain the frequency explanation, we perform a Fourier transformation of $e(\boldsymbol{x})$, resulting in the frequency explanation $\hat{e}(\boldsymbol{x}) \in \mathbb{C}^T$ with $\hat{E}$ for the entire set.

### 3.2 OBTAIN

The next step of RioT is to obtain user feedback on confounding factors. For an input $\boldsymbol{x}$, a user can mark parts that are confounded, resulting in a feedback mask $a(\boldsymbol{x}) \in \{0, 1\}^T$. In this binary mask, a 1 signals a potential confounder at this time step. Thereby, it is not necessary to have feedback for every sample of the dataset, as a mask $a(\boldsymbol{x}) = (0, \ldots, 0)^T$ corresponds to no feedback. Feedback can also be given on the frequency explanation in a similar manner, marking which elements in the frequency domain are potential confounders. The resulting feedback mask $\hat{a}(\boldsymbol{x}) = (\hat{a}(\boldsymbol{x})_{re}, \hat{a}(\boldsymbol{x})_{im})$ can be different for the real $\hat{a}(\boldsymbol{x})_{re} \in \{0, 1\}^T$ and imaginary part $\hat{a}(\boldsymbol{x})_{im} \in \{0, 1\}^T$. For the whole dataset, we then have spatial annotations $A$ and frequency annotations $\hat{A}$.

As the annotated feedback masks come from human experts, obtaining them can become costly. However, confounders often occur systematically, and it is thus possible to apply the same annotation mask to many samples. This can drastically reduce the number of annotations required in practice.

### 3.3 REVISE

The last step of RioT is integrating the feedback into the model. We apply the general idea of using a loss-based model revision (Schramowski et al., 2020; Ross et al., 2017; Stammer et al., 2020) based on the explanations and the annotation mask. Given the input data $(\mathcal{X}, \mathcal{Y})$, we define the original task (or right-answer) loss as $\mathcal{L}_{RA}(\mathcal{X}, \mathcal{Y})$. This loss measures the model performance and is the primary learning objective. To incorporate the feedback, we further use the right-reason loss $\mathcal{L}_{RR}(A, E)$. This loss aligns model explanations $E = \{e(\boldsymbol{x}) | \boldsymbol{x} \in \mathcal{X}\}$ and user feedback $A$ by penalizing the model for explanations in the annotated areas. To achieve model revision and a good task performance, both losses are combined, where $\lambda$ is a hyperparameter to balance both parts

of the combined loss $\mathcal{L}(\mathcal{X}, \mathcal{Y}, A, E) = \mathcal{L}_{\mathrm{RA}}(\mathcal{X}, \mathcal{Y}) + \lambda\mathcal{L}_{\mathrm{RR}}(A, E)$. Together, the combined loss simultaneously optimizes the primary training objective (e.g. accuracy) and feedback alignment.

**Time Domain Feedback.** Masking parts of the time domain as feedback is an easy way to mitigate spatially locatable confounders (Fig. 1, left). We use the explanations $E$ and annotations $A$ in the spatial version of the right-reason loss:

$$\mathcal{L}_{RR}^{sp}(A, E) = \frac{1}{D} \sum_{\boldsymbol{x}\in\mathcal{X}} (e(\boldsymbol{x}) * a(\boldsymbol{x}))^2 \qquad (3)$$

As the explanations and the feedback masks are element-wise multiplied, this loss minimizes the explanation values in marked parts of the input. This effectively trains the model to disregard the marked parts of the input for its computation. Thus, using the loss in Eq. 3 as right-reason component for the combined loss allows to effectively steer the model away from points or intervals in time.

**Frequency Domain Feedback.** However, feedback in the time domain is insufficient to handle every type of confounder. If the confounder is not locatable in time, giving spatial feedback cannot be used to revise the models' behavior. Therefore, we utilize explanations and feedback in the frequency domain to tackle confounders like in Fig. 1, (right). Given the frequency explanations $\hat{E}$ and annotations $\hat{A}$, the right-reason loss for the frequency domain is:

$$\mathcal{L}_{RR}^{fr}(\hat{A}, \hat{E}) = \frac{1}{D} \sum_{\boldsymbol{x}\in\mathcal{X}} \left( (\mathrm{Re}(\hat{e}(\boldsymbol{x})) * \hat{a}_{re}(\boldsymbol{x}))^2 + (\mathrm{Im}(\hat{e}(\boldsymbol{x})) * \hat{a}_{im}(\boldsymbol{x}))^2 \right) \qquad (4)$$

The Fourier transformation is invertible and differentiable, so we can backpropagate gradients to parameters directly from this loss. Intuitively, the frequency right-reason loss causes the masked frequency explanations of the model to be small while not affecting any specific point in time.

Depending on the problem at hand, it is possible to use RioT either in the spatial or frequency domain. Moreover, it is also possible to combine feedback in both domains, thus including two right-reason terms in the final loss. This results in two parameters $\lambda_1$ and $\lambda_2$ to balance the right-answer and both right-reason losses.

$$\mathcal{L}(\mathcal{X}, \mathcal{Y}, A, E) = \mathcal{L}_{\mathrm{RA}}(\mathcal{X}, \mathcal{Y}) + \lambda_1\mathcal{L}_{\mathrm{RR}}^{sp}(A, E) + \lambda_2\mathcal{L}_{\mathrm{RR}}^{fr}(\hat{A}, \hat{E}) \qquad (5)$$

It is important to note that giving feedback in the frequency domain allows a new form of model revision through XIL. Even if we effectively still apply masking in the frequency domain, the effect in the original input domain is entirely different. Masking out a single frequency affects all time points without preventing the model from looking at any of them. In general, having an invertible transformation from the input domain to a different representation allows to give feedback more flexible than before. The Fourier transformation is a prominent example but not the only possible choice for this. Using other transformations like wavelets (Graps, 1995), is also possible.

**Computational Costs.** Including RioT in the training of a model increases the computational cost. Computing the right reason loss term requires the computation of a mixed partial derivative: $\frac{\partial^2 f_\theta(x)}{\partial\theta\partial x}$. Even though this is a second-order derivative, it does not result in any substantial cost increases, as the second-order component of our loss can be formalized as a Hessian-vector product (cf. App. A.3), which is known to be fast to compute (Martens, 2010). We also observed this in our experimental evaluation, as even the naive implementation of our loss in PyTorch scales to large models.

## 4 EXPERIMENTAL EVALUATIONS

In this section, we investigate the effectiveness of RioT to mitigate confounders in time series classification and forecasting. Our evaluations include the potential of revising in the spatial domain (RioT$_{sp}$) and the frequency domain (RioT$_{freq}$), as well as both jointly.

### 4.1 EXPERIMENTAL SETUP

**Data.** We perform experiments on various datasets. For classification, we focus mainly on the UCR/UEA repository (Dau et al., 2018), which holds a wide variety of datasets for this task. The

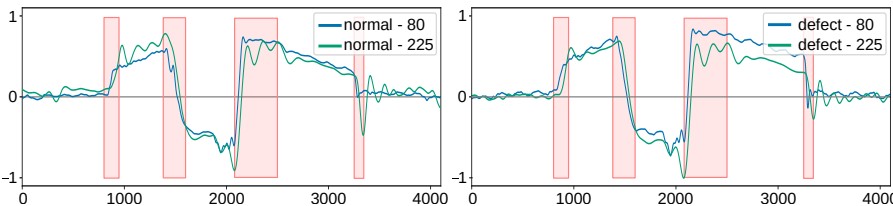

Figure 3: Samples of P2S with normal (left) and defect (right) setting at 80 and 225 strokes per minute. Areas that vary depending on the stroke rate and are considered confounding and marked red.

data originates from different domains, e.g., health records, industrial sensor data, and audio signals. We select all available datasets of a minimal size (cf. App. A.2), which results in FAULT DETECTION A, FORD A, FORD B, and SLEEP. We omit experiments on the very small datasets of UCR, as these are generally less suited for deep learning (Ismail Fawaz et al., 2020). We use the splits provided by the UCR archive. For time series forecasting, we evaluate on three popular datasets from the Darts repository (Herzen et al., 2022): ETTM1, ENERGY, and WEATHER. We split the data into training and test sets using a 70/30 ratio. The training set is further divided into training and validation subsets in an 80/20 ratio, resulting in an overall train/validation/test split of 56%/14%/30%. These datasets are sufficiently large, allowing us to investigate the impact of confounding behavior in isolation without the risk of overfitting. We standardize all datasets as suggested by Wu et al. (2021), i.e., rescaling the distribution of values to zero mean and a standard deviation of one.

**Production Press Sensor Data (P2S).** RioT aims to mitigate confounders in time series data. To assess our method, we need datasets with annotated real-world confounders. So far, there are no such datasets available. To fill this gap, we introduce PRODUCTION PRESS SENSOR DATA (P2S)[2], a dataset of sensor recordings with naturally occurring confounders. The sensor data comes from a high-speed press production line for metal parts, one of the sheet metal working industry's most economically significant processes. The task is to predict whether a run is defective based on the sensor data. The recordings include different production speeds, which, although not affecting part quality, influence process friction and applied forces. Fig. 3 shows samples recorded at different speeds from normal and defect runs, highlighting variations even within the same class. An expert identified regions in the time series that vary with production speed, potentially distracting models from relevant classification indicators, especially when no defect and normal runs of the same speed are in the training data. Thus, the run's speed is a confounder, challenging models to generalize beyond training. The default P2S setting includes normal and defect runs of different speeds, with an unconfounded setting of runs at the same speed. Further details on the dataset are available in App. B.

**Models.** For time series classification, we use the FCN model of Ma et al. (2023), with a slightly modified architecture for Sleep to achieve a better unconfounded performance (cf. App. A.1). Additionally, we use the OFA model by Zhou et al. (2023). For forecasting, we use the recently introduced TiDE model (Das et al., 2023), PatchTST (Nie et al., 2023) and NBEATS (Oreshkin et al., 2020) to highlight the applicability of our method to a variety of model classes.

**Metrics.** In our evaluations, we compare the performance of models on confounded and unconfounded datasets with and without RioT. For classification, we report balanced (multiclass) accuracy (ACC), and for forecasting the mean squared error (MSE). The corresponding mean absolute error (MAE) results can be found in App. A.5. We report average and standard deviation over 5 runs.

**Confounders.** To evaluate how well RioT can mitigate confounders in a more controlled setting, we add spatial (sp) or frequency (freq) shortcuts to the datasets from the UCR and Darts repositories. These confounders create spurious correlations between patterns and class labels or forecasting signals in the training data, but are absent in validation or test data. We generate an annotation mask based on the confounder area or frequency to simulate human feedback. More details on the confounders can be found in App. A.4.

---

[2] https://anonymous.4open.science/r/p2s

Table 1: **Applying RioT mitigates confounders in time series classification.** Performance before and after applying RioT for spatial (SP Conf) and frequency (Freq Conf) confounders. High training and low test accuracies indicate overfitting to the confounder, which RioT successfully mitigates. Unconfounded represents the ideal scenario where the model is not affected by any confounder.

| Model | Config (ACC ↑) | Fault Detection A | | FordA | | FordB | | Sleep | |
|---|---|---|---|---|---|---|---|---|---|
| | | Train | Test | Train | Test | Train | Test | Train | Test |
| FCN | Unconfounded | 0.99 ±0.00 | 0.99 ±0.00 | 0.92 ±0.01 | 0.91 ±0.00 | 0.93 ±0.00 | 0.76 ±0.01 | 0.68 ±0.00 | 0.62 ±0.00 |
| | SP Conf | **1.00** ±0.00 | 0.74 ±0.06 | **1.00** ±0.00 | 0.71 ±0.08 | **1.00** ±0.00 | 0.63 ±0.03 | **1.00** ±0.00 | 0.10 ±0.03 |
| | + RioT$_{sp}$ | 0.98 ±0.01 | **0.93** ±0.03 | 0.99 ±0.01 | **0.84** ±0.02 | 0.99 ±0.00 | **0.68** ±0.02 | 0.60 ±0.06 | **0.54** ±0.05 |
| | Freq Conf | **0.98** ±0.01 | 0.87 ±0.03 | **0.98** ±0.00 | 0.73 ±0.01 | **0.99** ±0.01 | 0.60 ±0.01 | **0.98** ±0.00 | 0.27 ±0.02 |
| | + RioT$_{freq}$ | 0.94 ±0.00 | **0.90** ±0.03 | 0.83 ±0.02 | **0.83** ±0.02 | 0.94 ±0.00 | **0.65** ±0.01 | 0.67 ±0.05 | **0.45** ±0.07 |
| OFA | Unconfounded | 1.00 ±0.00 | 0.98 ±0.02 | 0.92 ±0.01 | 0.87 ±0.04 | 0.95 ±0.01 | 0.70 ±0.04 | 0.69 ±0.00 | 0.64 ±0.01 |
| | SP Conf | **1.00** ±0.00 | 0.53 ±0.02 | **1.00** ±0.00 | 0.50 ±0.00 | **1.00** ±0.00 | 0.52 ±0.01 | **1.00** ±0.00 | 0.21 ±0.05 |
| | + RioT$_{sp}$ | 0.96 ±0.08 | **0.98** ±0.01 | 0.92 ±0.03 | **0.85** ±0.02 | 0.94 ±0.01 | **0.65** ±0.04 | 0.52 ±0.22 | **0.58** ±0.05 |
| | Freq Conf | **1.00** ±0.00 | 0.72 ±0.02 | **1.00** ±0.00 | 0.65 ±0.01 | 1.00 ±0.00 | 0.56 ±0.02 | **0.99** ±0.00 | 0.24 ±0.03 |
| | + RioT$_{freq}$ | 0.96 ±0.02 | **0.98** ±0.02 | 0.78 ±0.04 | **0.85** ±0.04 | **1.00** ±0.00 | **0.64** ±0.03 | 0.50 ±0.16 | **0.49** ±0.04 |

## 4.2 Evaluations

**Removing Confounders for Time Series Classification.** We evaluate the effectiveness of RioT (spatial: RioT$_{sp}$, frequency: RioT$_{freq}$) in addressing confounders in classification tasks by comparing balanced accuracy with and without RioT.

As shown in Tab. 1, without RioT, both FCN and OFA overfit to shortcuts, achieving ≈100% training accuracy, while having poor test performance. Applying RioT significantly improves test performance for both models across all datasets. In some cases, RioT even reaches the performance of the ideal reference (unconfounded) scenario as if there would be no confounder in the data. Even on FordB, where the drop in training-to-test performance of the reference indicates a distribution shift, RioT$_{sp}$ is still beneficial. Similarly, RioT$_{freq}$ enhances performance on frequency-confounded data, though the improvement is less pronounced for FCN on Ford B, suggesting essential frequency information is sometimes obscured by RioT$_{freq}$. In summary, RioT (both RioT$_{sp}$ and RioT$_{freq}$) successfully mitigates confounders, enhancing test generalization for FCN and OFA models.

**Removing Confounders for Time Series Forecasting.** Confounders are not exclusive to time series classification and can significantly impact other tasks, such as forecasting. In Tab. 2 we outline that spatial confounders cause models to overfit, but applying RioT$_{sp}$ reduces MSE across datasets, especially for Energy, where MSE drops by up to 56%. In the frequency-confounded setting, the training data includes a recurring Dirac impulse as a distracting confounder (cf. App. A.4 for details). RioT$_{freq}$ alleviates this distraction and improves the test performance significantly. For example, TiDE's test MSE on ETTM1 decreases by 14% compared to the confounded model.

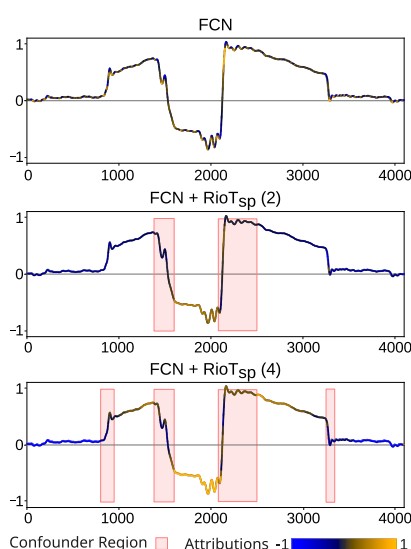

Figure 4: **Applying RioT lets the model ignore confounder areas.** While FCN primarily focuses on confounders, applying RioT with partial feedback (middle) or full feedback (bottom) causes the model to ignore the confounder and focus on the remainder of the input.

In general, RioT effectively addresses spatial and frequency confounders in forecasting tasks. Interestingly, for TiDE on the Energy dataset, the performance with RioT$_{freq}$ even surpasses the unconfounded model. Here, the added frequency acts as a form of data augmentation, enhancing model robustness.

Table 2: **RioT can successfully overcome confounders in time series forecasting.** MSE values (MAE values cf. Tab. 7) on the confounded training set and the unconfounded test set with Unconfounded being the ideal scenario where the model is not affected by any confounder.

| Model | Config (MSE ↓) | ETTM1 Train | ETTM1 Test | Energy Train | Energy Test | Weather Train | Weather Test |
|---|---|---|---|---|---|---|---|
| NBEATS | Unconfounded | 0.30 ±0.02 | 0.47 ±0.02 | 0.34 ±0.03 | 0.26 ±0.02 | 0.08 ±0.01 | 0.03 ±0.01 |
| | SP Conf | **0.24** ±0.01 | 0.55 ±0.01 | **0.33** ±0.03 | 0.94 ±0.02 | **0.09** ±0.01 | 0.16 ±0.04 |
| | + RioT$_{sp}$ | 0.30 ±0.01 | **0.50** ±0.01 | 0.45 ±0.03 | **0.58** ±0.01 | 0.11 ±0.01 | **0.09** ±0.02 |
| | Freq Conf | **0.30** ±0.02 | 0.46 ±0.01 | **0.33** ±0.04 | 0.36 ±0.04 | **0.11** ±0.02 | 0.32 ±0.09 |
| | + RioT$_{freq}$ | 0.31 ±0.02 | **0.45** ±0.01 | **0.33** ±0.04 | **0.34** ±0.04 | 0.81 ±0.48 | **0.17** ±0.01 |
| PatchTST | Unconfounded | 0.46 ±0.03 | 0.47 ±0.01 | 0.26 ±0.01 | 0.23 ±0.00 | 0.26 ±0.03 | 0.08 ±0.01 |
| | SP Conf | **0.40** ±0.02 | 0.55 ±0.01 | **0.29** ±0.01 | 0.96 ±0.03 | **0.20** ±0.03 | 0.19 ±0.01 |
| | + RioT$_{sp}$ | **0.40** ±0.03 | **0.53** ±0.01 | 0.44 ±0.00 | **0.45** ±0.01 | 0.55 ±0.20 | **0.14** ±0.01 |
| | Freq Conf | 0.45 ±0.03 | 0.91 ±0.16 | **0.04** ±0.00 | 0.53 ±0.05 | **0.63** ±0.09 | 0.24 ±0.04 |
| | + RioT$_{freq}$ | 0.91 ±0.07 | **0.66** ±0.04 | 0.71 ±0.10 | **0.38** ±0.06 | 0.96 ±0.02 | **0.17** ±0.00 |
| TiDE | Unconfounded | 0.27 ±0.01 | 0.47 ±0.01 | 0.27 ±0.01 | 0.26 ±0.02 | 0.25 ±0.02 | 0.03 ±0.00 |
| | SP Conf | **0.22** ±0.01 | 0.54 ±0.03 | **0.28** ±0.01 | 1.19 ±0.03 | **0.22** ±0.03 | 0.15 ±0.01 |
| | + RioT$_{sp}$ | 0.23 ±0.01 | **0.48** ±0.01 | 0.53 ±0.02 | **0.52** ±0.02 | 0.25 ±0.03 | **0.11** ±0.01 |
| | Freq Conf | **0.06** ±0.01 | 0.69 ±0.08 | **0.07** ±0.01 | 0.34 ±0.08 | **0.79** ±0.09 | 0.31 ±0.09 |
| | + RioT$_{freq}$ | 0.07 ±0.01 | **0.49** ±0.07 | **0.07** ±0.01 | **0.21** ±0.02 | 1.12 ±0.36 | **0.22** ±0.01 |

A similar behavior can also be observed for NBEATS and ETTM1, where the confounded setting actually improves the model slightly, and RioT even improves upon that.

**Removing Confounders in the Real-World.** So far, our experiments have demonstrated the ability to counteract confounders within controlled environments. However, real-world scenarios often have more complex confounder structures. Our new proposed dataset P2S presents such real-world conditions. The model explanations in Fig. 4 (top) reveal a focus on distinct regions of the sensor curve, specifically the two middle regions. With domain knowledge, it's clear that these regions shouldn't affect the model's output. By applying RioT, we can redirect the model's attention away from these regions. New model explanations highlight that the model still focuses on incorrect regions, which can be mitigated by extending the annotated area. In Tab. 3, the model performance (exemplarily with FCN) in these settings is presented. Without RioT, the model overfits to the confounder. the test performance improves already with partial feedback *(2)* and improves even more with full feedback *(4)*. These results highlight the effectiveness of RioT in real-world scenarios, where not all confounders are initially known.

**Removing Multiple Confounders at Once.** In the previous experiments, we illustrated that RioT is suitable for addressing individual confounding factors, whether spatial or frequency-based. Real-world time series data, however, often present a blend of multiple confounding factors that simultaneously may influence model performance. We thus investigate the impact of applying RioT to both spatial and frequency confounders simultaneously (cf. Tab. 4), exemplary using FCN and TiDE. When Sleep is confounded in both domains, FCN without RioT overfits and fails to generalize. Addressing only one confounder does not mitigate the effects, as the model adapts to the other. However, combining the respective feedback from both domains (RioT$_{freq,sp}$) significantly improves test performance, matching the frequency-confounded scenario (cf. Tab. 1). Tab. 4 (bottom) shows the impact of multiple confounders on the Energy dataset for forecasting. When faced with both spatial shortcut and noise confounders, the model overfits, indicated by lower training MSE. While applying either spatial or frequency feedback individually already shows some effect, utilizing both types of feedback simultaneously (RioT$_{freq,sp}$) results in the largest improvements, as both confounders are addressed. The performance gap between RioT$_{freq,sp}$ and the non-confounded model is more pronounced than in single confounder cases (cf. Tab. 2), suggesting a compounded challenge. Optimize the deconfounding process in highly complex data environments thus remains an important challenge.

**Handling Human Feedback.** Human feedback is a crucial component of RioT. To understand its impact, we conduct two ablation studies using the classification data set Fault Detection A and the

Table 3: **Applying RioT overcomes the confounder in P2S.** Performance on confounded train set and the unconfounded test set. FCN learns the train confounder, resulting in a drop in test performance. Applying RioT with partial feedback *(2)* already yields good improvements, while adding feedback on the full confounder area *(4)* is even better. Unconfounded is the ideal scenario, specifically curated so that there is no confounder.

| P2S (ACC ↑) | Train | Test |
|---|---|---|
| FCN$_{\text{Unconfounded}}$ | 0.97 ±0.01 | 0.95 ±0.01 |
| FCN$_{\text{sp}}$ | **0.99** ±0.01 | 0.66 ±0.14 |
| FCN$_{\text{sp}}$ + RioT$_{\text{sp}}$ (2) | 0.96 ±0.01 | 0.78 ±0.05 |
| FCN$_{\text{sp}}$ + RioT$_{\text{sp}}$ (4) | 0.95 ±0.01 | **0.82** ±0.06 |

Table 4: **RioT can combine spatial and frequency feedback.** If the data is confounded in time and frequency, RioT can combine feedback on both domains to mitigate confounders, superior to feedback on only one domain. Unconfounded represents the ideal scenario when the model is not affected by any confounder.

| Sleep (Classification ACC ↑) | Train | Test |
|---|---|---|
| FCN$_{\text{Unconfounded}}$ | 0.68 ±0.00 | 0.62 ±0.00 |
| FCN$_{\text{freq,sp}}$ | **1.00** ±0.00 | 0.10 ±0.04 |
| FCN$_{\text{freq,sp}}$ + RioT$_{\text{sp}}$ | 0.94 ±0.00 | 0.24 ±0.02 |
| FCN$_{\text{freq,sp}}$ + RioT$_{\text{freq}}$ | **1.00** ±0.00 | 0.04 ±0.00 |
| FCN$_{\text{freq,sp}}$ + RioT$_{\text{freq,sp}}$ | 0.47 ±0.00 | **0.48** ±0.03 |

| Energy (Forecasting MSE ↓) | Train | Test |
|---|---|---|
| TiDE$_{\text{Unconfounded}}$ | 0.28 ±0.01 | 0.26 ±0.02 |
| TiDE$_{\text{freq,sp}}$ | **0.16** ±0.01 | 0.74 ±0.02 |
| TiDE$_{\text{freq,sp}}$ + RioT$_{\text{sp}}$ | 0.20 ±0.01 | 0.61 ±0.02 |
| TiDE$_{\text{freq,sp}}$ + RioT$_{\text{freq}}$ | 0.22 ±0.01 | 0.55 ±0.02 |
| TiDE$_{\text{freq,sp}}$ + RioT$_{\text{freq,sp}}$ | 0.25 ±0.01 | **0.47** ±0.01 |

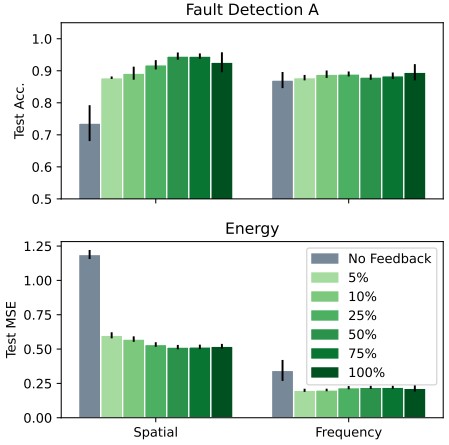

Figure 5: **RioT uses feedback efficiently.** Even with feedback on only a small percentage of the data, RioT can overcome confounders.

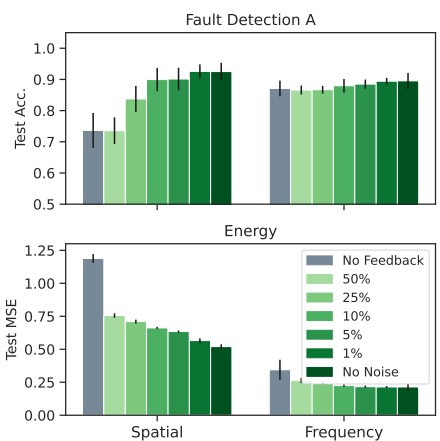

Figure 6: **RioT is robust against invalid feedback.** Even with some percentage of random feedback, RioT overcomes the confounders.

forecasting data set Energy. The first experiment examines the required amount of feedback, while the second assesses robustness to noisy feedback.

Recognizing that expert time is valuable and excessive feedback requests are impractical, our first experiment evaluates RioT 's performance when feedback is provided on only a portion of the dataset (Fig. 5). The findings reveal that full annotations are unnecessary. Even with minimal feedback, such as annotating just 5% of the samples, RioT significantly outperforms scenarios with no feedback.

While previous experiments assumed entirely accurate feedback, real-world applications often involve some degree of error. Therefore, we also test RioT 's resilience to increasing levels of incorrect feedback (Fig. 6). Instead of accurately marking confounding areas, random time steps or frequency components are incorrectly labeled as confounded. The results show that RioT maintains strong performance even with up to 10% invalid feedback, presenting only slight performance declines. In certain cases, like forecasting with spatial confounders, RioT can still achieve notable improvements despite high levels of feedback noise.

In summary, RioT effectively generalizes from small subsets of feedback and remains robust against a moderate amount of annotation noise. These qualities demonstrate that RioT is well-equipped to manage the practical challenges associated with human feedback.

**Limitations.** An important aspect of RioT is the human feedback provided in the Obtain step. Integrating human feedback into the model is a key advantage of RioT, but can also be a limitation. While we have shown that a small fraction of samples with annotations can be sufficient, and that annotations can be applied for many samples, they are still necessary for RioT. Additionally, like many other (explanatory) interactive learning methods, RioT assumes correct human feedback. Thus, considering possible repercussions of inaccurate feedback when applying RioT in practice is important. Another potential drawback of RioT are increased training costs. RioT requires computation of a mixed-partial derivative to optimize the model's explanation when using gradient-based attributions. While this affects training cost, the loss can be formulated as a Hessian-vector product, which is fast to compute in practice, making the additional overhead easy to manage.

## 5 CONCLUSION

In this work, we present Right on Time a method to mitigate confounding factors in time series data with the help of human feedback. By revising the model, RioT significantly diminishes the influence of these factors, steering the model to align with the correct reasons. Using popular time series models on several manually confounded datasets and the newly introduced, naturally confounded, real-world dataset P2S showcases that they are indeed subject to confounders. Our results, however, demonstrate that applying RioT to these models can mitigate confounders in the data. Furthermore, we have unveiled that addressing solely the time domain is insufficient for revising the model to focus on the correct reasons, which is why we extended our method beyond it. Feedback in the frequency domain provides an additional way to steer the model away from confounding factors and towards the right reasons. Extending the application of RioT to multivariate time series represents a logical next step, and exploring the integration of various explainer types is another promising direction. To further reduce the required human annotations, exploring the use of semi-automated feeback techniques, transfer learning or LLMs are potential next steps. Additionally, we aim to apply RioT, especially RioT$_{\text{freq}}$, to other modalities as well, offering a more nuanced approach to confounder mitigation. It should be noted that while our method shows potential in its current iteration, interpreting attributions in time series data remains a general challenge.

## 6 ETHICS STATEMENT

Our research aims to enhance the interpretability and reliability of time series models, with a focus on improving human interaction with time series models. By developing RioT, we prioritize guiding models toward correct reasoning, increasing transparency and trust in machine learning decisions. While human feedback plays a crucial role in refining these models, we acknowledge the potential for inaccuracies in the feedback. In production settings, it is crucial to consider these potential risks and implement appropriate safeguards to ensure responsible and reliable AI deployment.

## 7 REPRODUCIBILITY STATEMENT

To ensure the reproducibility of our research, we have made the code[3] and dataset[4] used in this work publicly available. Detailed instructions for running the experiments and replicating the results are provided in the repository, along with any dependencies and configurations required. The methodology is thoroughly described in the appendix App. A.1, outlining the steps implement and evaluate RioT. We encourage others to use these resources to validate and extend our findings.

---

[3]Code available at: `https://anonymous.4open.science/r/RioT`
[4]`https://anonymous.4open.science/r/p2s`

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

## A APPENDIX

### A.1 IMPLEMENTATION AND EXPERIMENTAL DETAILS

**Adaption of Integrated Gradients (IG).** A part of IG is a multiplication of the model gradient with the input itself, improving the explanation's quality (Shrikumar et al., 2017). However, this multiplication makes some implicit assumptions about the input format. In particular, it assumes that there are no inputs with negative values. Otherwise, multiplying the attribution score with a negative input would flip the attribution's sign, which is not desired. For images, this is unproblematic because they are always equal to or larger than zero. In time series, negative values can occur and normalization to make them all positive is not always suitable. To avoid this problem, we use only the input magnitude and not the input sign to compute the IG attributions.

**Computing Explanations.** To compute explanations with Integrated Gradients, we followed the common practice of using a baseline of zeros. The standard approach worked well in our experiments, so we did not explore other baseline choices in this work. For the implementation, we utilized the widely-used Captum[5] library, where we patched the `captum._utils.gradient.compute_gradients` function to allow for the propagation of the gradient with respect to the input to be propagated back into the parameters.

**Model Training and Hyperparameters.** To find suitable parameters for model training, we performed a hyperparameter search over batch size, learning rate, and the number of training epochs. We then used these parameters for all model trainings and evaluations, with and without RioT. In addition, we selected suitable $\lambda$ values for RioT with a hyperparameter selection on the respective validation sets. The exact values for the model training parameters and the $\lambda$ values can be found in the provided code[6].

To avoid model overfitting on the forecasting datasets, we performed shifted sampling with a window size of half the lookback window.

**Code.** For the experiments, we based our model implementations on the following repositories:

- FCN: `https://github.com/qianlima-lab/time-series-ptms/blob/master/model/tsm_model.py`
- OFA: `https://github.com/DAMO-DI-ML/NeurIPS2023-One-Fits-All/`
- NBEATS: `https://github.com/unit8co/darts/blob/master/darts/models/forecasting/nbeats.py`
- TiDE: `https://github.com/unit8co/darts/blob/master/darts/models/forecasting/tide_model.py`
- PatchTST: `https://github.com/awslabs/gluonts/tree/dev/src/gluonts/torch/model/patch_tst`

All experiments were executed using our Python 3.11 and PyTorch code, which is available in the provided code. To ensure reproducibility and consistency, we utilized Docker. Configurations and Python executables for all experiments are provided in the repository.

**Hardware.** To conduct our experiments, we utilized single GPUs from Nvidia DGX2 machines equipped with A100-40G and A100-80G graphics processing units.

By maintaining a consistent hardware setup and a controlled software environment, we aimed to ensure the reliability and reproducibility of our experimental results.

### A.2 UCR DATASET SELECTION

We focused our evaluation on a subset of UCR datasets with a minimum size. Our selection process was as follows: First, we discarded all multivariate datasets, as we only considered univariate data in this paper. Then we removed all datasets with time series of different length or missing values. We further excluded all datasets of the category *SIMULATED*, to avoid datasets which were synthetic

---

[5] `https://github.com/pytorch/captum`

[6] Code available at: `https://anonymous.4open.science/r/RioT`

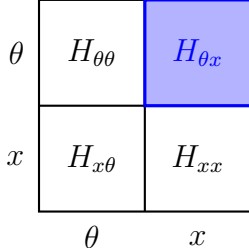

Figure 7: Illustration of the Hessian matrix with its respective sub-blocks. The mapping from $x$ into $\theta$ is highlighted in blue.

from the beginning. We furthermore considered only datasets with less than 10 classes, as having a per-class confounder on more than 10 classes would result in a very high number of different confounders, which would probably rarely happen. Besides these criteria, we discarded all datasets with less than 1000 training samples or a per sample length of less than 100, to avoid the small datasets of UCR, which leads to the resulting four datasets: Fault Detection A, Ford A, Ford B and Sleep.

### A.3 Computational Costs of RioT

Training a model with RioT induces additional computational costs. The right-reason term requires computations of additional gradients. Given a model $f_\theta(x)$, parameterized by $\theta$ and input $x$, then computing the right reason loss with a gradient-based explanation method requires the computation of the mixed-partial derivative $\frac{\partial^2 f_\theta(x)}{\partial\theta\partial x}$, as a gradient-based explanation includes the derivative $\frac{\partial f_\theta(x)}{\partial x}$. While this mixed partial derivative is a second order derivative, this does not substantially increase the computational costs of our method for two main reasons. First, we are never explicitly materializing the Hessian matrix. Second, the second-order component of our loss can be formulated as a Hessian-vector product:

$$\frac{\partial \mathcal{L}}{\partial \theta} = g + \frac{\lambda}{2} H_{\theta x}(e(x) - a(x)) \tag{6}$$

where $g = \frac{\partial \mathcal{L}_{\text{RA}}}{\partial \theta}$ is the partial derivative of the right answer loss and if $H$ is the full joint Hessian matrix of the loss with respect to $\theta$ and $x$, then $H_{\theta x}$ is the sub-block of this matrix mapping $x$ into $\theta$ (cf. Fig. 7), given by $H_{\theta x} = \frac{\partial^2 f_\theta(x)}{\partial\theta\partial x}$. Hessian-vector products are known to be fast to compute (Martens, 2010), enabling the right-reason loss computation to scale to large models and inputs.

### A.4 Details on Confounding Factors

In the datasets which are not P2S, we added synthetic confounders to evaluate the effectiveness of confounders. In the following, we provide details on the nature of these confounders in the four settings:

**Classification Spatial.** For classification datasets, spatial confounders are specific patterns for each class. The pattern is added to every sample of that class in the training data, resulting in a spurious correlation between the pattern and the class label. Specifically, we replace $T$ time steps with a sine wave according to:

$$confounder := \sin(t \cdot (2 + j)\pi)$$

while $t \in \{0, 1, \ldots, T\}$ and $j$ represents the class index, simulating a spurious correlation between the confounder and class index (Fig. 8).

**Classification Frequency.** Similar to the spatial case, frequency confounders for classification are specific patterns added to the entire series, altering all time steps by a small amount. The confounder is represented as a sine wave and is applied additively to the full sequence ($T = S$):

$$confounder := \sin(t \cdot (2 + j)\pi) \cdot A$$

where $A$ resembles the confounder amplitude.

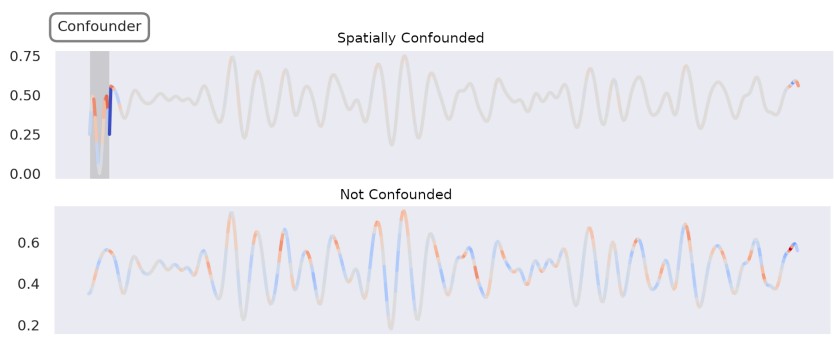

Figure 8: Example of the added spatial confounder in the Fault Detection dataset.

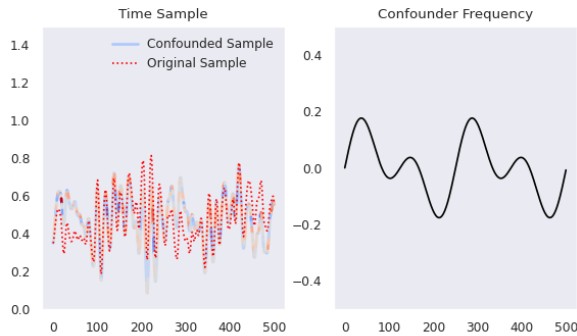

Figure 9: Example of an added frequency shortcut in the Fault Detection dataset.

**Forecasting Spatial.** For forecasting datasets, spatial confounders are shortcuts that act as the actual solution to the forecasting problem. Periodically, data from the time series is copied back in time. This "back-copy" is a shortcut for the forecast, as it resembles the time steps of the forecasting window. Due to the windowed sampling from the time series, this shortcut occurs at every second sample. The exact confounder formulation is outlined in the sketch below (Fig. 10), with an exemplary lookback length of 9, forecasting horizon of 3 and window stride of 6. This results in a shortcut confounder in samples 1 and 3 (marked red) and overlapping in sample 2 (marked orange). This can for example occur in scenarios such as data transmission. Transmission glitches, manifesting as packet losses or duplications, can subtly introduce irregularities into the time series data.

**Forecasting Frequency.** This setting differs from the previous shortcut confounders. The frequency confounder for forecasting is a recurring Dirac impulse with a certain frequency, added every $k$ time steps over the entire sequence (of length $S$), including the forecasting windows. This impulse is present throughout all of the training data, distracting the model from the real forecast. The confounder is present at all time steps: $i \in \{n \cdot k | n \in \mathbb{N} \wedge n \cdot k \leq S\}$ with a strength of $A$:

$$confounder := A \cdot \Delta_i$$

Such a confounder could, for example, occur when monitoring of water flow through a pipe in an assembly line. Suppose that during the capturing, a defect in one of the rollers adjacent to the water pipe induces a systematic stutter. This mechanical stutter, in turn, generates repeated impulses in the water flow sensor's readings. These impulses act as a systematic frequency confounder, which can negatively influence the performance of the forecasting model.

In conclusion, confounders are only present in the training data, not validation or test data. We generate an annotation mask based on the confounder area or frequency to simulate human feedback. This mask is applied to all confounded samples except in our feedback scaling experiment.

**1. Sample**

**Lookback**                                                                 **Horizon**

Unconfounded:  | 0 | 1 | 2 | 3 | 4 | 5 | 6 | 7 | 8 |     | 9 | 10 | 11 |

Confounded:    | 9 | 10 | 11 | 3 | 4 | 5 | 6 | 7 | 8 |     | 9 | 10 | 11 |

Feedback:      | 1 | 1 | 1 | 0 | 0 | 0 | 0 | 0 | 0 |

**2. Sample**

Unconfounded:  | 6 | 7 | 8 | 9 | 10 | 11 | 12 | 13 | 14 |     | 15 | 16 | 17 |

Confounded:    | 6 | 7 | 8 | 9 | 10 | 11 | 21 | 22 | 23 |     | 15 | 16 | 17 |

Feedback:      | 0 | 0 | 0 | 0 | 0 | 0 | 0 | 0 | 0 |

**3. Sample**

Unconfounded:  | 12 | 13 | 14 | 15 | 16 | 17 | 18 | 19 | 20 |     | 21 | 22 | 23 |

Confounded:    | 21 | 22 | 23 | 15 | 16 | 17 | 18 | 19 | 20 |     | 21 | 22 | 23 |

Feedback:      | 1 | 1 | 1 | 0 | 0 | 0 | 0 | 0 | 0 |

Confounder: ▢     Overlapping Confounder: ▢

Figure 10: Schematic overview of how the time series were confounded in the spatial forecasting experiments

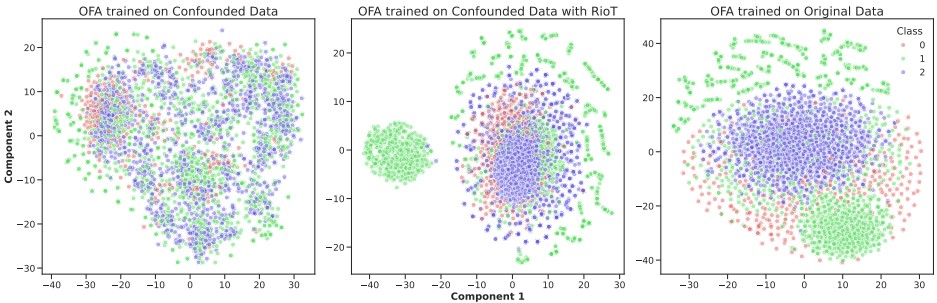

Figure 11: t-SNE plots of OFA encodings for Fault Detection A. The left plot shows a confounded model with minimal class separation. The middle plot shows a confounded model after RioT regularization, while the far right plot shows an unconfounded model with clear class separation. Both RioT-regularized and unconfounded model exhibit similar structures, highlighting the effectiveness of RioT.

## A.5 ADDITIONAL EXPERIMENTAL RESULTS

This section provides further insights into our experiments, covering both forecasting and classification tasks. Specifically, it showcases performance through various metrics such as MAE, MSE, and accuracy, qualitative insights about the influence of confounders, and explores different feedback configurations.

**Qualitative Insights into Encodings of a Confounded Model.** In Fig. 11, t-SNE plots outline the feature encodings of OFA trained on Fault Detection A under three different scenarios. The plot on the left represents a model trained on confounded data without any regularization, showing

Table 5: Feedback percentage for forecasting across all datasets, reported for the TiDE model. Corresponding to (test) results shown in Fig. 5, a higher percentage indicates more feedback, lower is better.

| Metric | Feedback | ETTM1 | | Energy | | Weather | |
|--------|----------|-------|------|--------|------|---------|------|
| | | Spatial | Freq | Spatial | Freq | Spatial | Freq |
| MAE ($\downarrow$) | 0% | 0.54 ±0.01 | 0.74 ±0.06 | 0.85 ±0.01 | 0.53 ±0.07 | 0.29 ±0.01 | 0.49 ±0.09 |
| | 5% | 0.52 ±0.00 | 0.63 ±0.03 | 0.62 ±0.01 | **0.40 ± 0.02** | 0.28 ±0.01 | 0.43 ±0.03 |
| | 10% | 0.52 ±0.00 | 0.63 ±0.03 | 0.61 ±0.01 | **0.40 ± 0.02** | 0.27 ±0.01 | 0.43 ±0.03 |
| | 25% | 0.52 ±0.00 | 0.63 ±0.03 | 0.58 ±0.01 | 0.41 ±0.01 | 0.25 ±0.01 | 0.43 ±0.04 |
| | 50% | 0.52 ±0.00 | 0.63 ±0.03 | **0.57 ± 0.01** | 0.41 ±0.01 | **0.24 ± 0.01** | 0.44 ±0.05 |
| | 75% | 0.52 ±0.01 | 0.63 ±0.03 | **0.57 ± 0.01** | 0.41 ±0.01 | **0.24 ± 0.01** | 0.45 ±0.06 |
| | 100% | **0.51 ± 0.01** | **0.60 ± 0.05** | 0.58 ±0.01 | **0.40 ± 0.03** | **0.24 ± 0.01** | **0.41 ± 0.02** |
| MSE ($\downarrow$) | 0% | 0.54 ±0.03 | 0.69 ±0.08 | 1.19 ±0.03 | 0.34 ±0.08 | 0.15 ±0.01 | 0.31 ±0.09 |
| | 5% | 0.54 ±0.01 | 0.52 ±0.03 | 0.60 ±0.02 | **0.20 ± 0.01** | 0.14 ±0.01 | 0.24 ±0.02 |
| | 10% | 0.53 ±0.01 | 0.52 ±0.03 | 0.57 ±0.02 | **0.20 ± 0.01** | 0.14 ±0.01 | 0.24 ±0.02 |
| | 25% | 0.53 ±0.01 | 0.52 ±0.03 | 0.53 ±0.02 | 0.22 ±0.01 | **0.11 ± 0.01** | 0.24 ±0.03 |
| | 50% | 0.53 ±0.01 | 0.52 ±0.03 | **0.51 ± 0.02** | 0.22 ±0.01 | **0.11 ± 0.01** | 0.25 ±0.04 |
| | 75% | 0.52 ±0.01 | 0.51 ±0.03 | 0.52 ±0.02 | 0.22 ±0.01 | **0.11 ± 0.01** | 0.26 ±0.05 |
| | 100% | **0.48 ± 0.01** | **0.49 ± 0.07** | 0.52 ±0.02 | 0.21 ±0.02 | **0.11 ± 0.01** | **0.22 ± 0.01** |

Table 6: Feedback percentage for classification across all datasets, reported for the FCN model. Corresponding to results shown in Fig. 5, a higher percentage indicates more feedback, higher is better.

| Feedback | Fault Detection A (ACC ↑) | | FordA (ACC ↑) | | FordB (ACC ↑) | | Sleep (ACC ↑) | |
|----------|---------|------|---------|------|---------|------|---------|------|
| | Spatial | Freq | Spatial | Freq | Spatial | Freq | Spatial | Freq |
| 0% | 0.74 ±0.06 | 0.87 ±0.03 | 0.71 ±0.08 | 0.73 ±0.01 | 0.63 ±0.03 | 0.60 ±0.01 | 0.10 ±0.03 | 0.27 ±0.02 |
| 5% | 0.88 ±0.00 | 0.88 ±0.01 | 0.81 ±0.03 | 0.80 ±0.03 | 0.66 ±0.03 | **0.66 ± 0.02** | 0.53 ±0.03 | **0.49 ± 0.00** |
| 10% | 0.89 ±0.02 | 0.89 ±0.01 | 0.82 ±0.04 | 0.79 ±0.02 | 0.66 ±0.03 | 0.64 ±0.03 | 0.48 ±0.09 | 0.48 ±0.02 |
| 25% | 0.92 ±0.01 | 0.89 ±0.01 | 0.83 ±0.02 | 0.78 ±0.01 | 0.67 ±0.02 | 0.65 ±0.01 | 0.49 ±0.08 | 0.42 ±0.08 |
| 50% | **0.95 ± 0.01** | 0.88 ±0.01 | 0.82 ±0.03 | 0.81 ±0.05 | 0.67 ±0.02 | 0.65 ±0.00 | **0.55 ± 0.03** | 0.44 ±0.07 |
| 75% | **0.95 ± 0.01** | 0.88 ±0.01 | 0.81 ±0.03 | 0.80 ±0.04 | 0.65 ±0.03 | 0.64 ±0.00 | 0.54 ±0.04 | 0.44 ±0.07 |
| 100% | 0.93 ±0.03 | **0.90 ± 0.03** | **0.84 ± 0.02** | **0.83 ± 0.02** | **0.68 ± 0.02** | 0.65 ±0.01 | 0.54 ±0.05 | 0.45 ±0.07 |

minimal class separation and no discernible structure, indicating poor feature representation caused by confounding factors. The middle plot depicts the same confounded model after applying RioT regularization, where class separation and improved structure emerge. The far right plot displays the eoncodings of a model trained on unconfounded data, with clear and distinct class clusters.

The qualitative insights are further supported by the scores presented in Tab. 1, which detail the corresponding classification performance of the presented models. The OFA model trained on confounded data achieves only ≈50% accuracy, while the RioT-regularized model regains nearly 100% accuracy, comparable to the unconfounded model. This improvement in accuracy aligns with the latent representations observed in the t-SNE plots, where RioT effectively steers the confounded model's structure to resemble that of the unconfounded data. These results highlight RioT's capability to mitigate confounding and restore robust model performance.

**Feedback Generalization.**: Tab. 6 and Tab. 5 detail provided feedback percentages for forecasting and classification across all datasets, respectively. These tables report the performance of the TIDE and FCN models, highlighting how different levels of feedback impact model outcomes on various datasets. Tab. 5 focuses on MAE and MSE for forecasting, while Tab. 6 presents ACC for classification.

**Removing Confounders for Time Series Forecasting.** Tab. 7 reports the MAE results for our forecasting experiment across different models, datasets and configurations. It emphasizes how well each model performs on both the confounded training set and after applying RioT, with the Unconfounded configuration representing the ideal scenario unaffected by confounders.

**Removing Multiple Confounders at Once.** Tab. 8 reports the MAE values and illustrates the effectiveness of combining spatial and frequency feedback via RioT for the TiDE model. The results demonstrate significant improvements in forecasting accuracy when integrating both feedback domains compared to using them separately.

Table 7: **RioT can successfully overcome confounders in time series forecasting.** MAE values on the confounded training set and the unconfounded test set with Unconfounded being the ideal scenario where the model is not affected by any confounder.

| Model | Config (MAE ↓) | ETTM1 | | Energy | | Weather | |
|-------|----------------|-------|------|--------|------|---------|------|
| | | Train | Test | Train | Test | Train | Test |
| NBEATS | Unconfounded | 0.39 ±0.01 | 0.48 ±0.01 | 0.44 ±0.02 | 0.38 ±0.01 | 0.21 ±0.01 | 0.12 ±0.01 |
| | SP Conf | **0.34** ±0.01 | 0.54 ±0.01 | **0.44** ±0.03 | 0.77 ±0.01 | **0.21** ±0.01 | 0.30 ±0.04 |
| | + RioT$_{sp}$ | 0.40 ±0.01 | **0.52** ±0.01 | 0.53 ±0.02 | **0.62** ±0.01 | 0.23 ±0.01 | **0.22** ±0.01 |
| | Freq Conf | **0.39** ±0.01 | 0.47 ±0.01 | **0.45** ±0.03 | 0.45 ±0.03 | **0.21** ±0.03 | 0.45 ±0.06 |
| | + RioT$_{freq}$ | 0.40 ±0.01 | **0.47** ±0.01 | **0.45** ±0.03 | **0.44** ±0.02 | 0.59 ±0.22 | **0.39** ±0.01 |
| PatchTST | Unconfounded | 0.50 ±0.01 | 0.49 ±0.01 | 0.39 ±0.00 | 0.38 ±0.01 | 0.38 ±0.03 | 0.18 ±0.00 |
| | SP Conf | **0.46** ±0.00 | 0.53 ±0.01 | **0.41** ±0.00 | 0.78 ±0.01 | **0.32** ±0.04 | 0.33 ±0.00 |
| | + RioT$_{sp}$ | **0.46** ±0.01 | **0.52** ±0.01 | 0.51 ±0.00 | **0.53** ±0.01 | 0.54 ±0.12 | **0.28** ±0.00 |
| | Freq Conf | **0.53** ±0.01 | 0.81 ±0.07 | **0.15** ±0.00 | 0.64 ±0.03 | **0.58** ±0.03 | 0.41 ±0.05 |
| | + RioT$_{freq}$ | 0.92 ±0.05 | **0.80** ±0.02 | 0.97 ±0.86 | **0.57** ±0.02 | 0.65 ±0.01 | **0.40** ±0.01 |
| TiDE | Unconfounded | 0.36 ±0.01 | 0.48 ±0.01 | 0.40 ±0.01 | 0.38 ±0.02 | 0.36 ±0.02 | 0.13 ±0.00 |
| | SP Conf | **0.32** ±0.01 | 0.54 ±0.01 | **0.40** ±0.01 | 0.85 ±0.01 | **0.32** ±0.03 | 0.29 ±0.01 |
| | + RioT$_{sp}$ | 0.34 ±0.01 | **0.51** ±0.01 | 0.57 ±0.01 | **0.58** ±0.01 | 0.35 ±0.03 | **0.24** ±0.01 |
| | Freq Conf | **0.18** ±0.01 | 0.74 ±0.06 | **0.18** ±0.01 | 0.53 ±0.07 | **0.65** ±0.05 | 0.49 ±0.09 |
| | + RioT$_{freq}$ | 0.19 ±0.01 | **0.60** ±0.05 | **0.18** ±0.01 | **0.40** ±0.03 | 0.79 ±0.16 | **0.41** ±0.02 |

Table 8: **RioT can combine spatial and frequency feedback.** MAE results when applying feedback in time and frequency with RioT. Combining both feedback domains is superior to feedback on only one of the domains. Reference values represent the ideal scenario when the model is not affected by any confounder (mean and std over 5 runs).

| Energy (MAE ↓) | Train | Test |
|----------------|-------|------|
| TiDE$_{Unconfounded}$ | 0.40 ±0.01 | 0.38 ±0.02 |
| TiDE$_{freq,sp}$ | **0.30** ±0.01 | 0.70 ±0.02 |
| TiDE$_{freq,sp}$ + RioT$_{sp}$ | 0.34 ±0.01 | 0.64 ±0.01 |
| TiDE$_{freq,sp}$ + RioT$_{freq}$ | 0.36 ±0.01 | 0.60 ±0.01 |
| TiDE$_{freq,sp}$ + RioT$_{freq,sp}$ | 0.39 ±0.01 | **0.55** ±0.01 |

**Early Stopping as Confounder Mitigation Baseline.** In this experiment, we compare the performance of a model with RioT to a model regularized via early stopping (which is decided on an unconfounded validation set). In that, we stop model training if there are no improvements in the validation set for several epochs in the hope that it thus does not overfit to the confounder. The results are presented in Tab. 9 for classification and Tab. 10 for forecasting. We can observe that early stopping can help in some instances to achieve performances similar to RioT (e.g. PatchTST with a frequency confounder or FCN with a spatial confounder). However, for the majority of cases the performance with early stopping is substantially lower than the performance with RioT, signaling that early stopping alone is not a sufficient approach to overcome confounders.

| Config | FCN, Train | FCN, Test | OFA, Train | OFA, Test |
|---|---|---|---|---|
| Reference | $0.99 \pm 0.00$ | $0.99 \pm 0.00$ | $1.00 \pm 0.00$ | $0.98 \pm 0.02$ |
| SP Conf | $1.00 \pm 0.00$ | $0.74 \pm 0.06$ | $1.00 \pm 0.00$ | $0.53 \pm 0.02$ |
| $\text{RioT}_{sp}$ | $0.98 \pm 0.01$ | $0.93 \pm 0.03$ | $0.96 \pm 0.08$ | $0.98 \pm 0.01$ |
| $\text{ES}_{sp}$ | $0.87 \pm 0.01$ | $0.91 \pm 0.03$ | $0.69 \pm 0.02$ | $0.67 \pm 0.04$ |
| Freq Conf | $0.98 \pm 0.01$ | $0.87 \pm 0.03$ | $1.00 \pm 0.00$ | $0.72 \pm 0.02$ |
| $\text{RioT}_{freq}$ | $0.94 \pm 0.00$ | $0.90 \pm 0.03$ | $0.96 \pm 0.02$ | $0.98 \pm 0.02$ |
| $\text{ES}_{freq}$ | $0.83 \pm 0.01$ | $0.86 \pm 0.02$ | $0.81 \pm 0.01$ | $0.75 \pm 0.02$ |

Table 9: Early stopping on classification datasets.

| Config | PatchTST, Train | PatchTST, Test | TiDE, Train | TiDE, Test |
|---|---|---|---|---|
| Reference | $0.26 \pm 0.01$ | $0.23 \pm 0.00$ | $0.27 \pm 0.01$ | $0.26 \pm 0.02$ |
| SP Conf | $0.29 \pm 0.01$ | $0.96 \pm 0.03$ | $0.28 \pm 0.01$ | $1.19 \pm 0.03$ |
| $\text{RioT}_{sp}$ | $0.44 \pm 0.00$ | $0.45 \pm 0.01$ | $0.53 \pm 0.02$ | $0.52 \pm 0.02$ |
| $\text{ES}_{sp}$ | $0.48 \pm 0.05$ | $0.68 \pm 0.03$ | $1.20 \pm 0.25$ | $0.81 \pm 0.08$ |
| Freq Conf | $0.04 \pm 0.00$ | $0.53 \pm 0.05$ | $0.07 \pm 0.01$ | $0.34 \pm 0.08$ |
| $\text{RioT}_{freq}$ | $0.71 \pm 0.10$ | $0.38 \pm 0.06$ | $0.07 \pm 0.01$ | $0.21 \pm 0.02$ |
| $\text{ES}_{freq}$ | $0.48 \pm 0.09$ | $0.49 \pm 0.08$ | $0.21 \pm 0.08$ | $0.36 \pm 0.09$ |

Table 10: Early stopping on forecasting datasets.

## B  CONFOUNDED DATASET FROM A HIGH-SPEED PROGRESSIVE TOOL

The presence of confounders is a common challenge in practical settings, affecting models in diverse ways. As the research community strives to identify and mitigate these issues, it becomes imperative to test our methodologies on datasets that mirror the complexities encountered in actual applications. However, for the time domain, datasets with explicitly labeled confounders are not present, highlighting the challenge of assessing model performance against the complex nature of practical confounding factors.

To bridge this gap, we introduce P2S, a dataset that represents a significant step forward by featuring explicitly identified confounders. This dataset originates from experimental work on a production line for deep-drawn sheet metal parts, employing a progressive die on a high-speed press. The sections below detail the experimental approach and the process of data collection.

### B.1  REAL-WORLD SETTING

The production of parts in multiple progressive forming stages using stamping, deep drawing and bending operations with progressive dies is generally one of the most economically significant manufacturing processes in the sheet metal working industry and enables the production of complex parts on short process routes with consistent quality. For the tests, a four-stage progressive die was used on a Bruderer BSTA 810-145 high-speed press with varied stroke speed. The strip material to be processed is fed into the progressive die by a BSV300 servo feed unit, linked to the cycle of the press, in the stroke movement while the tools are not engaged. The part to be produced remains permanently connected to the sheet strip throughout the entire production run. The stroke height of the tool is 63 mm and the material feed per stroke is 60 mm. The experimental setup with the progressive die set up on the high-speed press is shown in Fig. 12.

The four stages include a pilot punching stage, a round stamping stage, deep drawing and a cut-out stage. In the first stage, a 3 mm hole is punched in the metal strip. This hole is used by guide pins in the subsequent stages to position the metal strip. During the stroke movement, the pilot pin always engages in the pilot hole first, thus ensuring the positioning accuracy of the components. In the subsequent stage, a circular blank is cut into the sheet metal strip. This is necessary so that the part can be deep-drawn directly from the sheet metal strip. This is a round geometry that forms small

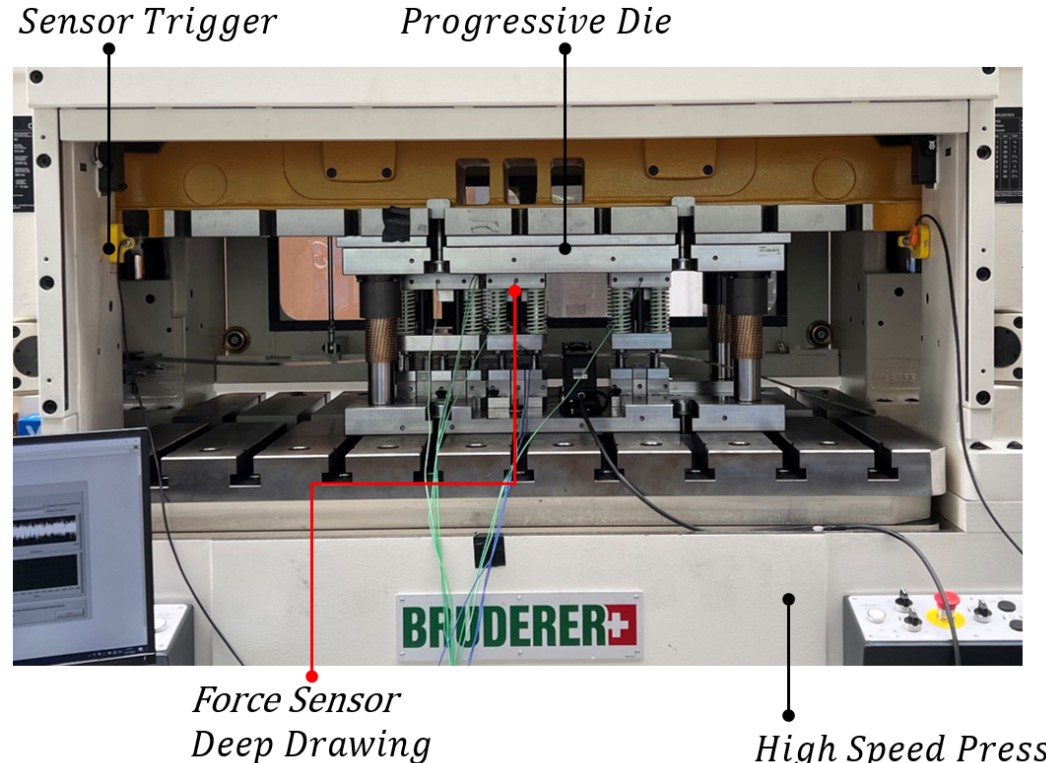

Figure 12: Experimental setup with high-speed press and tool as well as trigger for stroke-by-stroke recording of the data

arms in the subsequent deep-drawing step that hold the component on the metal strip. In the final stage, the component is then separated from the sheet metal strip and the process cycle is completed. The respective process steps are performed simultaneously so that each stage carries out its respective process with each stroke and therefore a part is produced with each stroke. Fig. 13 shows the upper tool unfolded and the forming stages associated with the respective steps on the continuous sheet metal strip.

## B.2 DATA COLLECTION

An indirect piezoelectric force sensor (Kistler 9240A) was integrated into the upper mould mounting plate of the deep-drawing stage for data acquisition. The sensor is located directly above the punch and records not only the indirect process force but also the blank holder forces which are applied by spring assemblies between the upper mounting plate and the blank holder plate. The data is recorded at a sampling frequency of 25 kHz. The material used is DC04 with a width of 50 mm and a thickness of 0.5 mm. The voltage signals from the sensors are digitised using a "CompactRIO" (NI cRIO 9047) with integrated NI 9215 measuring module (analogue voltage input $\pm$ 10 V). Data recording is started via an inductive proximity switch when the press ram passes below a defined stroke height during the stroke movement and is stopped again as it passes the inductive proximity switch during the return stroke movement. Due to the varying process speed caused by the stroke speeds that have been set, the recorded time series have a different number of data points. Further, there are slight variations in the length of the time series withing one experiment. For this reason, all time series are interpolated to a length of 4096 data points and we discard any time series that deviate considerably from the mean length of a run (i.e., threshold of 3). A total of 12 series of experiments, shown in Tab. 11, were carried out with production rates from 80 to 225 spm. To simulate a defect, the spring hardness of the blank holder was manipulated in the test series that were marked as *defect*. The manipulated experiments result in the component bursting and tearing during production. In a real production environment, this would lead directly to the parts being rejected.

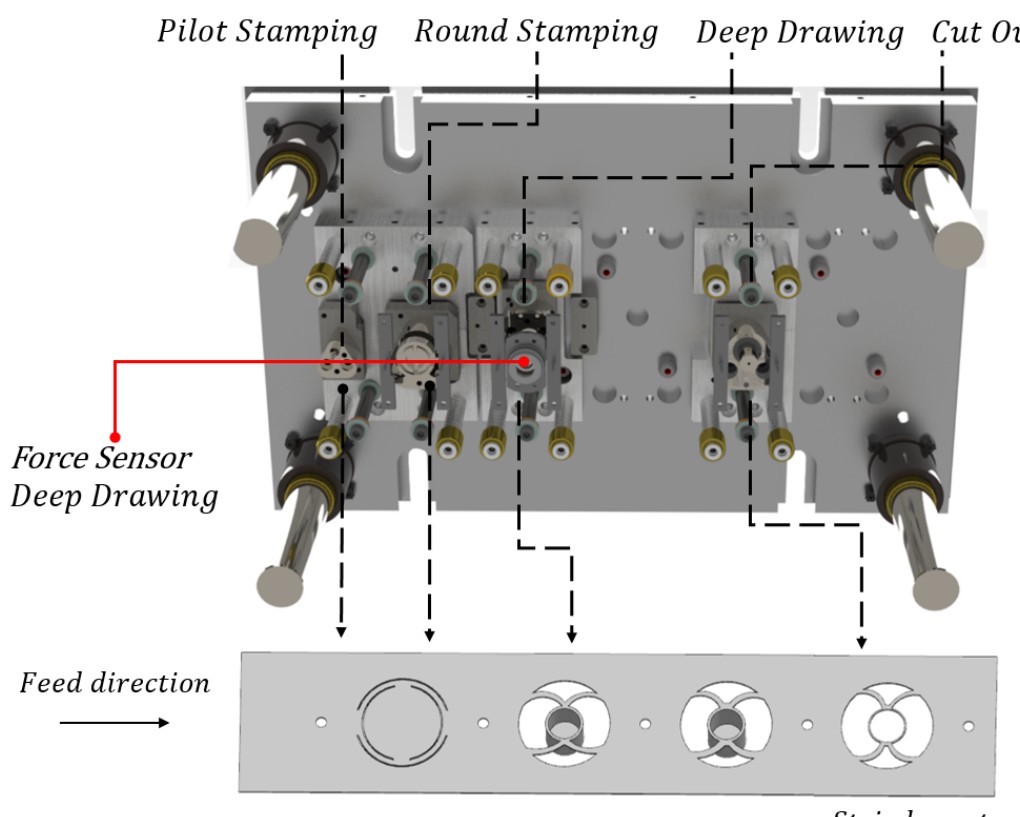

Figure 13: Upper tool unfolded and the forming stages associated with the respective steps on the passing sheet metal strip as well as the positions of the piezoelectric force sensors.

### B.3 DATA CHARACTERISTICS

Fig. 14 shows the progression of the time series recorded with the indirect force sensor. The force curve characterises the process cycle during a press stroke. The measurement is started by the trigger which is activated by the ram moving downwards. The downholer plates touch down at point A and press the strip material onto the die. Between point A and point B, the downholder springs are compressed, causing the applied force to increase linearly. The deep drawing process begins at point B. At point C, the press reaches its bottom dead centre and the reverse stroke begins so that the punch moves out of the material again. At point D, the deep-drawing punch is released from the material and now the hold-down springs relax linearly up to point E. At point E, the downholder plate lifts off again, the component is fed to the next process step and the process is complete.

### B.4 CONFOUNDERS

The presented dataset P2S is confounded by the speed with which the progressive tool is operated. The higher the stroke rate of the press, the more friction is occurring and the higher is the impact of the downholder plate. The differences can be observed in Fig. 3. Since we are aware of these physics-based confounders, we are able to annotate them in our dataset. As the process speed increases, the friction that occurs between the die and the material in the deep-drawing stage changes, as the frictional force is dependent on the frictional speed. This is particularly evident in the present case, as deep-drawing oils, which can optimize the friction condition, were not used in the experiments. The affected area from friction of the punch are in 1380 to 1600 (start of deep drawing) and 2080 to 2500 (end of deep drawing). In addition, the impulse of the downholder plate affecting the die increases due to the increased process dynamics. If the process speed is increased, the process force

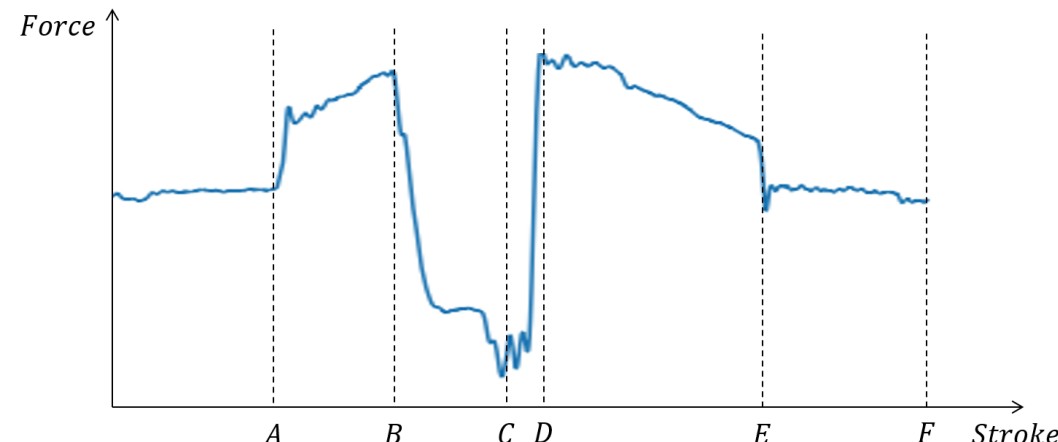

Figure 14: Force curve for one stroke. A) set down downholder plate B) start of deep drawing C) bottom dead centre D) deep drawing process completed E) downholder plates lift off F) measurement stops.

Table 11: Overview of conducted runs on the high-speed press with normal and defect states at different stroke rates.

| Experiment # | State | Stroke Rate | Samples |
| --- | --- | --- | --- |
| 1 | Normal | 80 | 193 |
| 2 | Normal | 100 | 193 |
| 3 | Normal | 150 | 189 |
| 4 | Normal | 175 | 198 |
| 5 | Normal | 200 | 194 |
| 6 | Normal | 225 | 188 |
| 7 | Defect | 80 | 149 |
| 8 | Defect | 100 | 193 |
| 9 | Defect | 150 | 188 |
| 10 | Defect | 175 | 196 |
| 11 | Defect | 200 | 193 |
| 12 | Defect | 225 | 190 |
| Total | | | 2264 |

also increases in the ranges of the time series from 800 to 950 (downholder plate sets down) and 3250 to 3550 (downholder plate lifts).

In the experiment setting of Tab. 3, the training data set is selected in such a way that the stroke rate correlates with the class label, i.e., there are only normal experiments with small stroke rates and defect ones with high stroke rate. Experiment 1, 2, 3, 10, 11, 12 are the training data and the remaining experiments are the test data. To get a unconfounded setting where the model is not affected by any confounder, we use normal and defect experiments with the same speed in training and respectively test data. This results in experiments 1, 3, 5, 7, 9, 11 in the training set and the remaining in the test set.

