# OpenReview forum: "Right on Time: Revising Time Series Models by Constraining their Explanations"
_ICLR.cc/2025/Conference — Submitted to ICLR 2025_

### Official Review · Reviewer_acPg · 2024-10-22

**Soundness:** 2
**Presentation:** 1
**Contribution:** 2
**Rating:** 3
**Confidence:** 4

**Summary:**

This paper proposed a method named Right on Time (RioT), which can enable human interactions with the time series model explanations in both time and frequency domains. The authors also created a confounded dataset named P2S from a real scenario and conducted detailed experiments on both synthetic datasets and this real-world datasets for evaluation.

**Strengths:**

* The idea of human-in-the-loop for removing confounding factors in time series modelling is interesting.
* The published confounded dataset will be beneficial to the research community.

**Weaknesses:**

* The proposed method seems to be not very practical in real world applications. Firstly, the proposed method relies on human annotations for the whole datasets, which will require huge labelling costs. Secondly, for many real-world applications, the confounders are often unknown, so that human labelling may not be trivial, especially in frequency domain.
* The proposed method assumes that all the confounders could be lablled by human experts. First, if all the condounders are known, it will be very convenient to apply causal learning based methods to avoid these confounding effects. Second, the proposed method penalizes the confounding factors by a penality term, which can only reduce their effects but cannot completely remove their effects. Directly removing the confounding factors from the input seems to be more effective than the proposed penality term in my opinion.
* The frequency domain explanation seems to be a trivial Fourier transformation of the time domain explanation according the descriptions in line 186/187. This means that the two types of explanations are essentially the same thing with different format, which may not be the optimal cases in my opinion. I am not sure if it would be better to directly learn frequency domain explanation independently, i.e., tranforming the input data into frequency domain and then applying IG for obtaining explanations.
* The presentation should be significantly improved. Here are a few examples:
    * line 34: in practice Geirhos et al. (2020) -> in practice (Geirhos et al., 2020)
    * line 124: casual analysis -> causal analysis
    * line 149: Given is a dataset - > Given a dataset

**Questions:**

For the major questions, please refer to my comments above.

Here are some additionl questions:
1. For Table 1, will the accuracy be the same as the unconfounded case if you directly remove all the confounders from the synthetic dataset? If so, a penality term will be unnecessary.
2. In line 200, you mentioned "apply the same annotation mask to many samples". How did you do that automatically?
3. Would it be possible to apply other unconfounding/causal learning/robust learning methods on the P2S datasest? It would be helpful to compare with other types of methods in addition to comparing different variants of the proposed method.

---

> ### Author Response · Authors · 2024-11-16
>
> Thank you for taking the time to review our manuscript. We appreciate your feedback and address each of your comments below.
>
> **(W1) RioT relies on costly human feedback:** We refer to our global response where we discuss this point, clarifying the amount of feedback required, handling of unknown confounders and confounders in the frequency domain.
>
> **(W2) Using Causal learning methods to remove confounders:**
> Knowledge of where the confounder occurs is not necessarily sufficient to apply causal learning methods. Generally, they have additional assumptions about, for example, the data generation process [1]. Additionally, RioT can mitigate confounders occurring within one variate, while typical causal time series deconfounding approaches only consider variates as a whole as confounded or relevant [2]. While knowledge about confounders can, in general, also be used for causal learning methods, that is not a trivial solution to the problem.
>
> **Removing confounded areas:**
> In cases where the confounder always occurs at the beginning of the time series, simply removing the time steps from all training and subsequent test examples could also be a solution. However, this strategy fails if the confounder’s position varies. In such scenarios, simply removing the confounder would also destroy the alignment of time steps between samples and make learning quite difficult. Applying RioT to a model enables it to ignore confounders without these alignment issues. Similar difficulties arise when attempting to remove confounders from multivariate data. In the case of a frequency confounder, this approach is generally not possible anymore, as there is no specific time step that corresponds to the confounder and could be removed.
>
> **(W3) Frequency explanations:**
> While the frequency explanation is "only" a transformed version of the spatial explanation, this change in representation is quite important. Regularizing models in the spatial domain is limited to spatially separable confounders. When a confounder is, for example, a sine wave frequency that overlaps with the data, spatial feedback is not helpful anymore, as the confounder occurs at all points in time. It has been shown that models are prone to frequency shortcuts/confounders even when only confronted with spatial input [3], and using the presented frequency explanation is a suitable tool to mitigate this. Further, transforming the input to the frequency domain and then applying IG is essentially equivalent, as the FFT is a linear transformation that a single matrix multiplication can represent.
> Nevertheless, we agree that directly learning frequency explanations is a very interesting direction for future research, and such explanations could then directly be incorporated in RioT as well.
>
> **(W4) Presentation:**
> Thanks for pointing these typos out. We checked the paper and improved the writing. We kindly ask whether the reviewer has any other specific points where the presentation of the paper should be improved, particularly in the light that the other reviewers found the paper to be clearly structured and well-written.
>
> **(Q1)** See answer to W2
>
> **(Q2) Reapplying Annotations to other samples:** In many cases, the same annotation can be reapplied to other samples as well. When examining P2S, the domain expert identified the run speed as a potential confounder for the outcome of the experiments and then annotated a few (5 to 10) samples with feedback masks. As the confounder always occurs at a specific phase of the run, the feedback masks could then be used for all samples, requiring no further annotations.
>
> **(Q3) Alternative Approaches for P2S** So far, other methods to mitigate confounders (for example, through causal learning) on time series data rely on a setting where a full variate of the time series can be considered confounded and the other variates contain the relevant information. As the confounder and the relevant information for the classification are part of the same variate in P2S, they cannot be applied easily. Other confounder mitigation work has so far been concentrated on the visual and textual domain without considering time series. The scarcity of alternative methods to tackle this problem is one of the important motivations for introducing RioT.

---

> > ### Author Response · Authors · 2024-11-16
> >
> > [1] Tobias Hatt and Stefan Feuerriegel. Sequential deconfounding for causal inference with unobserved confounders. In Proceedings of the Conference on Causal Learning and Reasoning (CLeaR), 2024.
> >
> > [2] Defu Cao, James Enouen, Yujing Wang, Xiangchen Song, Chuizheng Meng, Hao Niu, and Yan Liu. Estimating treatment effects from irregular time series observations with hidden confounders. In Proceedings of the AAAI Conference on Artificial Intelligence (AAAI), 2023.
> >
> > [3] Shunxin Wang, Christoph Brune, Raymond Veldhuis, and Nicola Strisciuglio. 2023. DFM-X: Augmentation by leveraging prior knowledge of shortcut learning. In Proceedings of the IEEE/CVF International Conference on Computer Vision. 129–138.

---

> > ### Comment · Reviewer_acPg · 2024-11-26
> > **Response to the author's rebuttal**
> >
> > I appreciate the authors' responses to my questions. However, I am not fully convinced by the rebuttal:
> >
> > 1. The removal of confounding data is a widely accepted practice within the research community to enhance the reliability of time series modeling. I disagree with the authors' claim that "this strategy fails if the confounder’s position varies. In such scenarios, simply removing the confounder would also destroy the alignment of time steps between samples and make learning quite difficult." Artifact removal is a common preprocessing step in time series modeling, and there are typically no significant difficulties in learning time series models with some missing values. In many real-world scenarios, such as ICU measurements, missing data occurs naturally. For frequency confounders, in scenarios like physiological signal processing, it is standard practice to remove data at 50/60Hz to mitigate artifacts caused by utility frequency or to discard data outside a specific frequency range.
> >
> > 2. Relying solely on FFT for frequency domain analysis is insufficient for time series modeling. In specific domains, such as speech analysis, the Mel-Spectrogram is more commonly used than FFT for frequency analysis and interpretation. Given that the Mel-Spectrogram is non-linear, I do not believe the authors can apply IG as stated with FFT.
> >
> > 3. For causal learning methods, confounders are usually harder to model within the data generation process. If human experts can label all the confounders, it will be easier for these experts to define the data generation process.
> >
> > Overall, I am still concerned about the validity of the claims, the limitation of technical choices, and the insufficient comparison/discussion of related works.

---

> ### Author Response · Authors · 2024-11-28
>
> We thank the reviewer for the response and acknowledge the remaining concerns.
>
> **1. Removal of confounding data:** We agree that in certain scenarios, removing confounded data can increase the robustness of time series models. As the reviewer rightly points out, this is a common preprocessing step, extending beyond time series data. However, removing (or imputing) confounded data has still to be done carefully, as this process can inadvertently introduce new confounders, which a model may similarly exploit.
>
> To illustrate this, we conducted two additional experiments where we compare a model trained without any confounder mitigation, one using imputation of annotated data, and RioT leveraging the same annotations. The results demonstrate that simply imputing the confounder (which even only occurs in one class) does not improve model performance. Instead, the model tends to use the imputed data as a shortcut to solve the task. Similarly, in the frequency setting, we remove the annotated frequencies from the data. However, this also introduces new shortcuts in the data, as the removal of said frequencies impacts the data from both classes differently. In contrast, RioT effectively addresses the confounder in both scenarios by training the model to disregard the confounder in a more generalized manner.
>
> In summary, while we agree with the reviewer that removing confounded data is a crucial preprocessing step, RioT does not aim to replace this process. Rather, RioT complements it by teaching the model to generalize and robustly ignore confounders, particularly in more complex cases.
>
> | FordA, Spatial Confounder | FCN, Train     | FCN, Test       |
> |:-----------------------------|:----------------------|:----------------------|
> | Reference                    | $0.92 \pm \tiny 0.01$ | $0.91 \pm \tiny 0.00$ |
> | SP Conf                      | $1.00 \pm \tiny 0.00$ | $0.71 \pm \tiny 0.01$ |
> | $\text{Imputation}_\text{sp}$ | $1.00 \pm \tiny 0.00$ | $0.72 \pm \tiny 0.01$ |
> | $\text{RioT}_\text{sp}$       | $0.99 \pm \tiny 0.01$ | $0.84 \pm \tiny 0.01$ |
>
>
>
> | FordA, Freq Confounder | FCN, Train     | FCN, Test       |
> |:-------------------------|:----------------------|:----------------------|
> | Reference                | $0.92 \pm \tiny 0.01$ | $0.91 \pm \tiny 0.00$ |
> | Freq Conf                | $0.98 \pm \tiny 0.00$ | $0.73 \pm \tiny 0.01$ |
> | $\text{Cut}_\text{freq}$  | $0.97 \pm \tiny 0.00$ | $0.73 \pm \tiny 0.01$ |
> | $\text{RioT}_\text{freq}$ | $0.83 \pm \tiny 0.02$ | $0.83 \pm \tiny 0.02$ |
>
>
> **2. Other frequency representations:** We agree that the FFT is not a one-size-fits-all solution for analyzing the frequency domain. Since the Mel-Spectrogram is only approximately invertible, using this transformation can introduce additional challenges. While it is possible to generate Mel-Spectrogram-based explanations in a manner similar to FFT-based explanations, applying them for model optimization is less straightforward.
> So far, RioT can easily incorporate transformations that are invertible, like STFT or Wavelets alongside FFT, but extending this approach to other types of transformation or exploring alternate methods to generate frequency explanations are important steps for future work.
>
>
> **3. Causal learning methods:** Generally, knowledge about confounders does not imply (full) knowledge about the data generation process, as confounders represent only a subset of the overall data generation mechanism. However, when experts can provide the necessary information, causal learning, and deconfounding methods can be applied, provided the confounder is represented as a separate variate.
>
> Alternatively, it is also possible to employ causal discovery to obtain this information (e.g., with https://jakobrunge.github.io/tigramite/). These methods, however, rely on their own assumptions, such as Causal Sufficiency or Faithfulness [1], which have to be considered when applying them in real-world scenarios.
>
> Overall, we agree that when the required assumptions are satisfied, applying causal learning methods to mitigate confounders is a good approach. Importantly, we emphasize that RioT can be used without relying on these specific assumptions.
>
> Hopefully, this could clarify the remaining concerns. If not, we are happy to answer any follow-up questions.
>
> [1] Jakob Runge et al., Detecting and quantifying causal associations in large nonlinear time series datasets.Sci. Adv.5,eaau4996(2019).DOI:10.1126/sciadv.aau4996

---

> > ### Comment · Reviewer_acPg · 2024-12-02
> > **Responses to the author's new results**
> >
> > I am not convinced by the authors' claim that removing (or imputing) confounded data will introduce "new shortcuts in the data". How can "removing confounded data" introduce new shortcuts in the data? If this is indeed the case, it may only indicate that you are applying the wrong imputation method.
> >
> > Also, the results showed obvious overfitting. For example, in the first table, both SP conf and Imputation_sp have 100% training accuracy but very low test accuracy. Therefore, using early stopping or other commonly used techniques for addressing overfitting in the experiments should be reaonable as suggested in other reviews.
> >
> > I checked the code and did not find any specific implementation for addressing the overfitting issue on the baselines. This may be problematic, because any machine learning model cannnot perform well without proper model training or hyperparameter tuning. Therefore, It could be the case that the major performance improvements come from using regularization (the proposed penality term) vs. not using regularization (the baselines using stardard cross entropy / MSE loss) in the experiments.
> >
> > BTW, in the above experiment, are you using the same dataset for the early stopping experiment?

---

> > > ### Author Response · Authors · 2024-12-03
> > >
> > > When removing the confounder (via imputation), a new confounder can be introduced, as the imputed data is a new pattern itself. Even in cases where random noise is imputed, this can still be a pattern distinct from the remainder of the time series. In this example, we used mean imputation, and despite the imputed mean being different for each sample, the pattern still remains a confounder. While we acknowledge that there may exist methodologies that omit this problem, the most commonly used methods do not mitigate the problem.
> > >
> > > As we discussed in the general response above, when confronted with confounders in the data, models tend to overfit to these confounders. However, simply using early stopping does not solve this problem, as confounders are generally already learned early during training [1], which our experiment above confirms. As early stopping does not improve the performance of the confounded models, we did not include this in the setup. Besides that, we did optimize the base model's hyperparameters as well, as described in the paper. We can add the hyperparameter tuning scripts to the repository.
> > >
> > > The early stopping experiment in the general response above is on the dataset Fault Detection, but the same holds for FordA; here is the performance of the base model with early stopping in this setting:
> > >
> > >
> > > | FordA, Spatial Confounder | FCN, Train     | FCN, Test       |
> > > |:-----------------------------|:----------------------|:----------------------|
> > > | Reference                    | $0.92 \pm \tiny 0.01$ | $0.91 \pm \tiny 0.00$ |
> > > | SP Conf                       | $1.00 \pm \tiny 0.00$ | $0.71 \pm \tiny 0.01$ |
> > > | $\text{Imputation}_\text{sp}$ | $1.00 \pm \tiny 0.00$ | $0.72 \pm \tiny 0.01$ |
> > > | $\text{ES}_\text{sp}$         | $0.99 \pm \tiny 0.00$ | $0.78 \pm \tiny 0.02$ |
> > > | $\text{RioT}_\text{sp}$       | $0.99 \pm \tiny 0.01$ | $0.84 \pm \tiny 0.01$ |
> > >
> > >
> > > | FordA, Freq Confounder | FCN, Train     | FCN, Test       |
> > > |:-------------------------|:----------------------|:----------------------|
> > > | Reference                | $0.92 \pm \tiny 0.01$ | $0.91 \pm \tiny 0.00$ |
> > > | Freq Conf                | $0.98 \pm \tiny 0.00$ | $0.73 \pm \tiny 0.01$ |
> > > | $\text{Cut}_\text{freq}$  | $0.97 \pm \tiny 0.00$ | $0.73 \pm \tiny 0.01$ |
> > > | $\text{ES}_\text{freq}$  | $0.84 \pm \tiny 0.02$ | $0.80 \pm \tiny 0.02$ |
> > > | $\text{RioT}_\text{freq}$ | $0.83 \pm \tiny 0.02$ | $0.83 \pm \tiny 0.02$ |
> > >
> > >
> > >
> > > [1] Yu Yang, Eric Gan, Gintare Karolina Dziugaite, and Baharan Mirzasoleiman. 2024. Identifying spurious biases early in training through the lens of simplicity bias. In International Conference on Artificial Intelligence and Statistics. PMLR, 2953–2961.

---

### Official Review · Reviewer_4ai3 · 2024-10-29

**Soundness:** 2
**Presentation:** 2
**Contribution:** 2
**Rating:** 5
**Confidence:** 4

**Summary:**

The paper introduces RioT, a method to mitigate confounders in time series analysis. The authors adapt established explanatory interactive learning (XIL) methods for time series analysis. XIL includes human-in-the-loop approaches that provide feedback on model explanations, particularly penalising model decisions based on confounding factors such as artifacts and noise. Evaluations across various architectures and downstream applications indicate that integrating human feedback into the training process may help reduce confounding factors in time series analysis.

**Strengths:**

1. The paper is well structured.

2. The paper is clearly written.

3. The authors conduct various experiments including different model architectures and downstream applications to evaluate their method.

4. The authors provide their code to support reproducibility.

**Weaknesses:**

1. Table 1 to Table 4 as well as the statement "we evaluate on [...] popular datasets [...] with 70% / 30% train / test splits" (ll. 268-269) indicate that the authors did not use a validation set, i.e. tuned their models on the test set. The authors should follow good practice and adhere to established procedures, e.g. as introduced by [1] for forecasting (train/val/test split of 60/20/20 for ETTm1, and 70/10/20 for Energy and Weather).

2. The proposed approach requires human feedback for each downstream application, which is expensive as domain knowledge is required. Furthermore, the authors claim that "RioT is well-equipped to manage the practical challenges associated with human feedback" (ll 475-476). However, such human-in-the-loop approaches are vulnerable to adversarial attacks, i.e. intentionally poisoned feedback, which the authors do not address in their work.

3. The approach focuses on uni-variate time series analysis. Since the authors do not explain how to extend their method to multi-variate time series, its applicability to real-world scenario remains limited. Additionally, the authors further constrain their approach by excluding "datasets with time series of different length or missing values" (l. 752), focusing on "datasets with less than 10 classes" (l. 769), and omitting "all datasets with less than 1000 training samples" (ll. 771-772) from their analysis. These constraints, especially the first and last, rule out applications in domains with limited data availability, e.g. medicine.

4. The authors use integrated gradients [2] to provide explanations $e(x)$, assuming that labels are available, e.g. for classification tasks, and of good quality. For forecasting tasks, pseudo-labels are created by computing an explanation for each time step $w$ in the forcasting horizon $W$ and creating a single representative explanation using global average pooling over all $W$ explanations, which requires further investigations to rule out points of failure.

5. The methodology section requires clarification, as the authors introduce variables, including $\bar{x}$, $\tilde{x}$, and $\alpha$ in Equation (1), without providing definitions.

6. The reproducibility is not fully supported, as the authors did not describe how they tuned the models for the experiments.

[1] Zhou et al. "Informer: Beyond efficient transformer for long sequence time-series forecasting." AAAI (2021).

[2] Sundararajan et al. "Axiomatic attribution for deep networks." ICML (2017).

**Questions:**

1. How to provide feedback for medical domains and domains which might contain non-trivial confounders? Does the approach translate well across domains such that feedback might not be required for specific applications?

2. How to evaluate whether the human feedback is of good quality? Is there a way to quantitatively measure feedback quality other than training the model with the respective human feedback and eventually evaluating the downstream performance?

3. How is the human-in-the-loop supposed to identify the confounders in the real and imaginary part within the frequency domain? Could the authors further elaborate on this and provide visual examples for clarity?

4. The model needs to be trained to derive the explanations using integrated gradients. How many iterations of training, i.e. explanations $e(x)$, and successive feedback, i.e. $a(x)$, are required to achieve a sufficiently good model? How does the downstream performance evolve as the number of iterations increase?

5. Could overfitting on confounders be mitigated by carefully selecting the validation set? As the authors assume that "confounders [...] are absent in [the] test set" (ll. 309-310), why not construct a validation set with non-confounded samples and use an early stopping criterion on the validation metric to prevent overfitting? This experimental setup would demonstrate how the proposed RioT performs in comparison to cheap early stopping techniques, that do not require human feedback.

6. The authors investigate fully supervised settings. Have they considered to analyse whether pre-trained models, i.e. models that are trained using self-supervised methods, are less prone to overfitting on confounders?

7. Do the authors "report average and standard deviation over 5 runs" (l. 305) across different seeds set during fine-tuning?

8. The authors "add spatial (sp) or frequency (freq) shortcuts to the datasets from the UCR and Darts repositories" (l. 308). For visualisation purposes, could they also provide examples of representative data samples before and after adding these shortcuts?

9. Could the authors add a qualitative analysis of the learned latent space to visualise how confounders may influence the model prediction?

---

> ### Author Response · Authors · 2024-11-16
>
> Thank you for your thoughtful review and constructive feedback on our paper. We address each of your comments and clarify points in the paper accordingly.
>
> **(W1) Validation Set Usage:** We apologize for the lack of clarity regarding our data split procedure. For forecasting we utilized a train/validation/test split of 56%/14%/30%, where the "train" portion was further divided into training and validation sets using an 80/20 split.
> We relied on the official test splits provided for datasets from the UCR repository (i.e., for classification tasks). In addition, we also provide train/test splits for P2S, where we also use 20% of the original training set as validation.
>
> Our optimization process is, of course, based on the validation set and not the test set. We have clarified this point in the manuscript and provided additional details about this setup.
>
> **(W2.1) Requirement of human feedback:** We provide a detailed discussion of this aspect in the general response above.
>
> **(W2.2) Adversarial Attacks:** We agree that adversarial attacks present an important challenge in human-in-the-loop systems. However, the scope of this work primarily focuses on establishing the feasibility of human feedback to mitigate confounders rather than performing a comprehensive adversarial analysis. Addressing adversarial attacks in these interactive methods, particularly in time series contexts, is indeed a critical area for future research. We clarified this now in more detail in our limitations section.
>
> **(W3.1) Extension to Multivariate Time Series:** Extending RioT to multivariate time series is both feasible and straightforward within our proposed framework. For brevity and to maintain controlled experimental conditions, this paper focuses on univariate series. To apply RioT to multivariate time series, an explanation method such as integrated gradients can be employed to provide insights across multiple variables. The feedback mechanism extends naturally to multivariate data by expanding the feedback mask dimensions to match the number of variables. Alternatively, it is also possible to mask a variate entirely if it is a potential confounder. This straightforward adjustment allows RioT to adapt seamlessly to multivariate settings without substantially modifying the core methodology.
>
> **(W3.2) Dataset Constraints and Generalizability:** While we selected specific datasets to ensure controlled experimental conditions, especially for adding synthetic confounders, this does not imply RioT is limited to these scenarios. As long as model explanations and feedback masks can be given for a time series, RioT can also be applied to that time series. This can also include settings with missing values or varying lengths, as IG can be applied in both scenarios. However, we agree that training in settings with very small amounts of data containing confounders can generally be challenging due to the tendency of deep learning models to overfit when training data is scarce [1]. However, this is a general problem in these circumstances and is not limited to RioT.
>
>
> **(W4) Integrated Gradients for Forecasting**:
> Per se, there is no technical difference between applying integrated gradients for classification or regression tasks, instead of using the class output neuron as a target, the regression output neuron is used (cf. [2]).
> What we add as a further adaption to enable IG for forecasting is the aggregation over the different neurons in the forecasting window. We believe that such an aggregation is reasonable, as we are more interested in the overall global features of the time series rather than the exact timestep details. Thus, despite its simplicity, aggregation serves as a reasonable proxy. However, we acknowledge that explanation and attribution methods for time series forecasting are still open research topics and encourage continued exploration in this field.
>
> **(W5) Clarification on Methodology and Notation:** Thank you for your remark, we apologize for this oversight. $\bar{{x}}$ is the baseline for integrated gradients. The method integrates the gradient along the path from baseline to the input value. Thereby, $\alpha$ is the integration variable, and $\tilde{{x}}$ is the temporary point on the path used for integration. We now introduce all missing variables in Equation (1).
>
> **(W6) Reproducibility and Model Tuning:** We tuned our hyperparameters using Optuna, and all training and setup scripts are included in the provided code repository to support reproducibility. Details are given in Appendix A1

---

> > ### Author Response · Authors · 2024-11-16
> >
> > **(Q1) Feedback in Complex Domains and Domain Translation:** Although mitigating non-trivial confounders can be challenging, our experiments demonstrate that even partial confounder annotations help reduce confounding effects. Generally, confounders and the resulting feedback is often specific to the domains, so translating feedback between different application domains may indeed pose challenges.
> >
> > **(Q2) Quality Assessment of Human Feedback:** This is an interesting question, and we agree that evaluating feedback quality is essential. As mentioned, model performance remains a primary proxy metric for feedback quality, but exploring other metrics, such as feedback alignment, could be valuable.
> >
> > **(Q3) Identifying Confounders in the Frequency Domain:** To clarify, identifying confounders can depend on the domain; for example, in audio data, the frequency domain is a natural perspective for human inspection [3]. In this domain, confounders such as background noise can be detected more naturally.
> >
> > **(Q4) Number of Training Iterations for Confounder Mitigation:** Generally, a single iteration of RioT can be sufficient. However, when the confounder is only partially known and annotated, multiple iterations can improve the outcomes, where inspecting explanations after one explanation can lead to refined feedback masks with RioT. For example, in the case of P2S, the final result was achieved with two RioT iterations (the final and partial results are shown in Tab. 3 and Fig. 4).
> >
> > **(Q5) Overfitting Mitigation Using Validation Sets:** This is an interesting suggestion. We evaluated how well early stopping can help in these situations, the results and the discussion are presented in the global response above.
> >
> > **(Q6) Analysis of Pre-trained Models and Self-supervised Settings:** While our work focused on fully supervised settings, exploring pre-trained models and self-supervised approaches to analyze resilience to confounders is indeed an interesting future direction. Work in other domains [4] has already shown that these training paradigms are not safe from confounders as well. We added this as a potential extension for RioT in our future work.
> >
> > **(Q7) Reporting Variance Over Multiple Runs:** Yes, we report the average and standard deviation across five runs with different seeds. For clarity, we added this detail to the experimental setup.
> >
> > **(Q8) Visual Examples of Data Samples with Added Shortcuts:** We appreciate the suggestion to include visual examples. We added representative data samples before and after adding spatial or frequency shortcuts to enhance transparency and interpretability to the appendix (A.3).
> >
> > **(Q9) Qualitative Analysis of Learned Latent Space:** We agree that visualizing the learned latent space could provide additional insights. While this was beyond the primary scope, we performed a small experiment exploring how confounders influence the latent space of a model with and without RioT. We added this to the appendix (A.5).
> >
> > [1] Hassan Ismail Fawaz, Benjamin Lucas, Germain Forestier, Charlotte Pelletier, Daniel F Schmidt, Jonathan Weber, Geoffrey I Webb, Lhassane Idoumghar, Pierre-Alain Muller, and François Petitjean. _Inceptiontime: Finding alexnet for time series classification._ Data Mining and Knowledge Discovery (DMKD), 34(6):1936–1962, 2020.
> >
> > [2] Alam, Md Shafiul, et al. "Applications of Integrated Gradients in Credit Risk Modeling." 2022 IEEE International Conference on Big Data (Big Data). IEEE, 2022.
> >
> > [3] Johanna Vielhaben, Sebastian Lapuschkin, Grégoire Montavon, and Wojciech Samek. Explainable AI for Time Series via Virtual Inspection Layers. ArXiv:2303.06365, 2023
> >
> > [4] Ryan Steed, Swetasudha Panda, Ari Kobren, and Michael Wick. 2022. Upstream mitigation is not all you need: Testing the bias transfer hypothesis in pre-trained language models. In Proceedings of the 60th Annual Meeting of the Association for Computational Linguistics (Volume 1: Long Papers). 3524–3542

---

> > > ### Author Response · Authors · 2024-11-27
> > >
> > > Dear Reviewer,
> > >
> > > The paper revision phase is approaching its conclusion. We hope that our responses and updates have fully addressed your concerns. If you have any additional comments, we would be happy to address them before the rebuttal period ends.
> > >
> > > Best regards,
> > > The Authors

---

> ### Comment · Reviewer_4ai3 · 2024-12-03
> **Reviewer Response**
>
> Thank you very much for the rebuttal, I have carefully read the authors responses and still have some concerns.
>
> ---
> **(W1)** Why not follow the splitting procedures established in time series analysis [1][2][3][4][5][6][7]?
>
> **(W2.2)** Unfortunately, a discussion on adversarial attacks is not included in the manuscript.
>
> **(W3.1)** I am sure that the methodology can be extended to a multivariate setting, however, I am uncertain whether it is feasible to provide human feedback on time series with a large number of variates?
>
> **(Q6)** It is unfortunate that the manuscript does not include self-supervised baselines pre-trained on large time series corpora, such as [7][9][10][11][12][13][14], as I believe a thorough investigation whether such pre-trained models are more robust against confounders would improve the quality of the study.
>
> **(Q9)** The authors provide t-SNE plots, which are known to be sensitive to parameter tuning. Therefore, a latent space analysis using principal component analysis would be the fairer approach.
>
> ---
> [1] Zhou, H. et al. "Informer: Beyond efficient transformer for long sequence time-series forecasting." AAAI Conference on Artificial Intelligence (AAAI). 2021.
>
> [2] Wu, H. et al. "TimesNet: Temporal 2D-Variation Modeling for General Time Series Analysis." International Conference on Learning Representations (ICLR). 2022.
>
> [3] Zhang, X. et al. “Self-supervised contrastive pre-training for time series via time-frequency consistency.” Advances in Neural Information Processing Systems (NeurIPS). 2022.
>
> [4] Liu, Y. et al. "iTransformer: Inverted Transformers Are Effective for Time Series Forecasting." International Conference on Learning Representations (ICLR). 2023.
>
> [5] Dong, J. et al. “SimMTM: A simple pre-training framework for masked time-series modeling.” Advances in Neural Information Processing Systems (NeurIPS). 2023.
>
> [6] Nie, Y. et al. “A time series is worth 64 words: Long-term forecasting with transformers.” International Conference on Learning Representations (ICLR). 2023.
>
> [7] Gao, S. et al. "UniTS: A unified multi-task time series model." Advances in Neural Information Processing Systems (NeurIPS). 2024.
>
> [8] Max Planck Institute for Biogeochemistry. “Weather station.” 2024.  URL: https://www.bgc-jena.mpg.de/wetter/.
>
> [9] Jin, M. et al. "Time-LLM: Time Series Forecasting by Reprogramming Large Language Models." International Conference on Learning Representations (ICLR). 2023.
>
> [10] Zhou, T. et al. "One fits all: Power general time series analysis by pretrained lm." Advances in Neural Information Processing Systems (NeurIPS). 2024.
>
> [11] Goswami, M. et al. "MOMENT: A Family of Open Time-series Foundation Models." International Conference on Machine Learning (ICML). 2024.
>
> [12] Woo, G. et al. "Unified Training of Universal Time Series Forecasting Transformers." International Conference on Machine Learning (ICML). 2024.
>
> [13] Yang, C. et al. “Biot: Biosignal transformer for cross-data learning in the wild.” Advances in Neural Information Processing Systems (NeurIPS). 2024.
>
> [14] Jiang, W. et al. “Large brain model for learning generic representations with tremendous EEG data in BCI.” International Conference on Learning Representations (ICLR). 2024.

---

> > ### Author Response · Authors · 2024-12-03
> >
> > We thank the reviewer for their response and are happy that we could clarify most of the concerns. In the following, we answer the remaining questions:
> >
> >
> > **(W1)** The works the reviewer mentioned are purely forecasting-based methods that utilize the https://github.com/thuml/Time-Series-Library, while we utilize Darts for loading our forecasting data. At the time of setting up our experiments, we were not aware of this library, and in hindsight, it would also be a viable option. As other works also rely on darts for their forecasting data ([1]), we believe this should not make any substantial difference. Additionally, as we do not compare models for raw performance but rather on the robustness against confounders in the data, our setup (independent of splits) further differs from the presented works.
> >
> >
> > **(W2)** You are right; we apparently forgot to add this part to the revised version. We update this for the camera ready version. For reference, the new limitation section is the following:
> >
> > "
> > An important aspect of \rrt is the human feedback provided in the Obtain step. Integrating human feedback into the model is a key advantage of RioT, but can also be a limitation. While we have shown that a small fraction of samples with annotations can be sufficient, and that annotations can be applied for many samples, they are still necessary for RioT. Additionally, like many other (explanatory) interactive learning methods, RioT assumes correct human feedback. Thus, considering possible repercussions of inaccurate feedback when applying RioT in practice is important. **In particular, it is also essential to investigate the potential of adversarial attacks which could be performed via poisoned feedback.**
> > Another potential drawback of RioT are increased training costs. RioT requires computation of a mixed-partial derivative to optimize the model’s explanation when using gradient-based attributions. While this affects training cost, the loss can be formulated as a Hessian-vector product, which is fast to compute in practice, making the additional overhead easy to manage.
> > "
> >
> > **(W3.1)** We agree that giving detailed feedback in a multivariate setting might be difficult. To alleviate this, there are multiple ways to give feedback in a multivariate setting that do not require fine-grained annotations:
> > - _Case 1: There is a potential shortcut at these time steps_: In this case, one can give feedback on these time steps and apply it to all variates simultaneously. While this also covers variates, which potentially do not contain the confounder, giving feedback is substantially simpler and can outweigh this.
> > - _Case 2: This variate contains potential confounders_: In this case, it is also possible to simply mark a full variate as confounded and generate a feedback mask for all time steps automatically.
> >
> > These are two exemplary and simple ways to give feedback in a multivariate setting as well, without requiring more detailed annotations. Nevertheless, we agree that investigating further ways for efficient human feedback in multivariate settings is an important direction for future research.
> >
> > **(Q6)** We agree that this is an interesting direction and will consider this in our next steps.
> >
> > **(Q9)** t-SNE is a commonly used way to analyze latent spaces, but we agree that one has to take the outcome with a grain of salt.
> > However, we can ensure that we did not "tune" any parameter and just used the default parameters from sklearn.
> > In the following, we conducted this analysis with PCA as well (Please refer to the uploaded plot: https://ibb.co/P4GWKz4). The conclusions that can be drawn are similar for both methods, but we will add the PCA plots to the camera-ready version as well.
> >
> >
> > We hope that this clarifies all remaining questions of the reviewer and are happy if they reconsider their score.
> > Best, the authors
> >
> >
> > [1] Gruver, Nate, et al. "Large language models are zero-shot time series forecasters." Spotlight NeurIPS(2024). (https://github.com/ngruver/llmtime)

---

### Official Review · Reviewer_aQ3G · 2024-11-02

**Soundness:** 3
**Presentation:** 2
**Contribution:** 3
**Rating:** 6
**Confidence:** 4

**Summary:**

The paper titled introduces a novel method named RioT for mitigating confounding factors in time series models. The authors emphasize the importance of addressing 'Clever-Hans' moments in time series models where spurious correlations may compromise model performance. The key contributions are:
1. Introduction of a new real-world dataset P2S containing annotated confounders from industrial sensor data.
2. Proposal of RioT, a feedback-based technique that operates across both time and frequency domains to mitigate confounders.
3. Empirical evidence showcasing that RioT improves the generalization of time series models across multiple datasets and scenarios.

**Strengths:**

1. High practical relevance: The focus on confounders in industrial sensor data highlights the method's real-world applicability.
2. Novel dual-domain feedback: Operating across both time and frequency domains is an innovative extension to existing XIL paradigms.
3. Empirical rigor: The experiments span across multiple datasets, models, and configurations, adding robustness to the findings.

**Weaknesses:**

1. Dependency on human feedback: The method requires expert annotations, which may be costly and challenging to acquire at scale.
2. Training cost: Incorporating feedback through gradient-based explanations adds computational overhead.
3. Limited exploration of feedback noise impact: Although robustness tests are performed, more exploration of noisy annotations could be useful to assess real-world viability.

**Questions:**

1. In Figure 1, it is unclear how annotations on the pretraining data are handled. Can the authors clarify if the feedback on pretraining is incorporated into final model updates?
2. Could you elaborate on any instances within which incorporating frequency domain feedback negatively affected performance? It would be particularly insightful to explore scenarios where the frequency domain annotations might conflict with those in the time domain.
3. Given the increased training cost, what strategies can practitioners use to balance the feedback quality and computational efficiency?

---

> ### Author Response · Authors · 2024-11-16
>
> Thank you for your review and constructive feedback on our work. We are encouraged by your positive assessment of the practical relevance of RioT, the dual feedback approach, and its empirical robustness. We appreciate the opportunity to address your concerns and clarify some key points.
>
> **(W1) Dependency on Human Feedback:** We provide a detailed discussion of this aspect in the general response above.
>
> **(W2) Training Cost:** While RioT's gradient-based feedback does introduce computational overhead, the actual computational effort remains manageable. In particular, even a naive implementation of RioT in PyTorch scales to large, pretrained models like OFA.
>
> **(W3) Limited Exploration of Feedback Noise:** We agree that evaluations with noisy and potentially incorrect feedback are important for all interactive learning methods. To start addressing this concern, we performed robustness tests regarding noisy feedback, where RioT achieved promising results. Nevertheless, we agree that more detailed evaluations on the robustness of RioT and XIL methods in general to noisy feedback are essential steps in future research in this field.
>
> **(Q1) The incorporation of Feedback:** Figure 1 illustrates the application of RioT to a confounded time series model. First, the model undergoes standard training, followed by an inspection of its explanations. Relevant annotations are collected from these explanations, and the model is subsequently retrained, this time optimized using both the target labels and the annotated explanation masks.
> While we do not explicitly consider pretrained models in this work, previous work has shown that explanation-based regularization can also be applied to already confounded models to correct them without the need for full retraining [1]
>
> **(Q2) Negative Impact of Frequency Domain Feedback:** In Fig. 6, we investigated how noisy feedback influences the model's performance, this included noisy feedback on the frequency domain and highlights that RioT is quite robust to noisy frequency feedback.
> In our experiments, we did not encounter situations where the frequency feedback did contradict the spatial feedback or vice versa. While such cases are, in principle, imaginable, they are not that likely to occur, as feedback in the frequency domain has a different impact on the model than feedback in the spatial domain due to the different representations.
>
> **(Q3) Balancing Feedback Quality and Computational Efficiency:** In cases where training costs are of high interest, the "switch XIL on" [1] strategy could be employed. At first, training happens without RioT, and then at a later part of the training process, RioT is used.
>
> [1] Felix Friedrich, Wolfgang Stammer, Patrick Schramowski, and Kristian Kersting. _A typology for exploring the mitigation of shortcut behaviour._ Nature Machine Intelligence, 5(3):319–330, 2023a.

---

> > ### Comment · Reviewer_aQ3G · 2024-11-25
> > **Thank you for your response.**
> >
> > The rebuttal is not very informative, as it mostly reiterates the claims made in the paper. It somehow addressed some of my questions.
> >
> > I appreciate the idea of leveraging human annotation to improve interpretability. Given this positive aspect, I will raise my score to 6.

---

> > > ### Author Response · Authors · 2024-11-25
> > >
> > > Thank you for your feedback and for raising your score. We appreciate your recognition of the value of leveraging human annotation for interpretability. While we understand your concern about the rebuttal reiterating points from the paper, we believed we had addressed all your questions. Could you kindly specify which aspects remain unclear or require further elaboration? We would be happy to provide more detailed responses.
> > >
> > > PS: We noticed that the score has not been updated on OpenReview yet. Would you mind checking this?

---

### Official Review · Reviewer_EVUz · 2024-11-03

**Soundness:** 4
**Presentation:** 3
**Contribution:** 3
**Rating:** 6
**Confidence:** 3

**Summary:**

The paper proposes a right for the right reason method (RRR) for time series data, which they refer to as Right On Time RioT. Similar to the RRR method, the method involves first extracting explanations for the model's predictions. They use IG for explanations, and then, using human feedback, they add a loss to penalize the model for looking at the incorrect locations. Since they do this for time series, they propose adding two losses, one for the time domain and one for the frequency domain. RioT reduces the model's reliance on confounding factors during training. The paper introduces a new dataset called PRODUCTION PRESS SENSOR DATA (P2S), which captures sensor readings from an industrial high-speed press in the sheet metal working industry. This dataset is characterized by naturally occurring confounders that can lead to inaccurate predictions when used for training models to detect production faults. P2S provides explicitly annotated confounders, P2S can be used for testing and comparing methods to mitigate the impact of confounders on real-world data.

## RioT

1- *Explain* The paper proposes using IG to extract model explanations for classification problems. The paper uses a modified version of IG with the input magnitude and not the input sign to compute the IG attribution equation 1; for forecasting, the paper uses the average over the forecasting window as an explanation, as shown in equation 2. To get explanations in the frequency domain, the paper transforms the explanations from the time domain to the frequency domain using the Fourier transformation.

2- *Obtain* The paper assumes the user provides annotations of cofounders, resulting in a mask {0,1} mask where 1 represents the presence of a confounder.

3-*Revise* An additional loss term is added to the objective that penalizes the model for looking at the wrong location. This is done by multiplying the explanation produced by the IG with the annotation mask. They propose adding a loss term for both frequency and time domains.

## Experiments

# Datasets

- The paper introduces P2S, which is annotated with confounders. It is a classification dataset. In this dataset, the paper refers to the run speed as a confounder, and experts highlight regions in the time series where there is a difference in the time series due to the difference in the run speed, not in the label itself.

- The paper evaluates UCR/UEA datasets to test whether RioT can help mitigate confounders by adding spatial or frequency shortcuts to the datasets. These confounders result in false correlations between patterns and class labels or forecasting signals within the training data, which do not appear in the validation or test data. An annotation mask is created to simulate human feedback based on the confounder's region or frequency.

# Models

- For classification, they use RioT on FCN and OFA models.

- For regression, they evaluate adding RioT to the TiDE, PatchTST, and NBEATS models.

# Results

- For classification: The paper showed that adding RioT helps improve the accuracy when the data has spurious correlations present in the training dataset but not in the test data across 4 datasets in both spatial and frequency domains.

- For forecasting: The paper showed that adding RioT for confounding datasets reduces MSE in some cases, giving MSE even better the the unconfounded datasets.

- On P2S: They showed that even with partial human feedback, RioT could help the model focus on the correct regions in the input data, resulting in better overall accuracy.

- Multiple confounders: The paper investigates when confounders exist in both frequency and spatial domains; it shows that using the aggregate loss functions with both frequency and spatial feedback gives the best results in this case.

- Human Feedback: The paper shows that even with a small amount of feedback, 5% of the original dataset significantly boosts accuracy. The paper also shows RioT is robust to noisy feedback. They showed that with even 50% noisy feedback, using RioT can still result in some performance gains.

**Strengths:**

## Originality

- The application of RRR methods to time series data is novel.
- Introducing two RRR loss terms for spatial and time domains is novel.
- The P2S dataset is a new contribution, featuring confounder annotations that can be utilized for future method evaluations.
- The unique approach to adding confounders to the UCR/UEA datasets is original; this dataset can serve a similar purpose to DomainBed in image analysis (https://github.com/facebookresearch/DomainBed) for future model evaluations.

## Quality

- The application of RRR in this context is well-justified.
- The paper presents robust empirical results.
- It explores various scenarios where feedback was either incomplete or incorrect, demonstrating that RioT remains effective in such cases.
- The paper is well-written and easy to understand.

## Significance

- With the introduction of the new P2S dataset and the proposed setup for UCR/UEA, this work is quite significant. It provides valuable resources that the machine learning community can use and build upon.

**Weaknesses:**

## Novelty:
-  RioT itself is not novel, it is just applying RRR to time series data.
## Clarity:
- Some minor clarity issues; please see the question section.

**Questions:**

- What is $\bar{x}$ in equation 1? if it's the reference point, please clarify and discuss how you choose the reference point for time series data.

- In equation 2, how is $e'_i(x)$ calculated?

- In line 249, why is the second-order derivative required? IG traditionally uses first-order derivatives.

- In Figure 4, it is difficult to see the attribution colors.

- In the experiment "Removing Multiple Confounders at Once," are the masks (that mimic human feedback) given in both spatial and frequency domains? i.e., different masks for different domains? If this is not the case, I am not sure why applying RRR in both frequency and time domains will necessarily help. Since one is the transformation of the other. What is your intuition as to why this might be the case?

---

> ### Author Response · Authors · 2024-11-16
>
> Thank you for your review and valuable feedback on our paper. We are pleased to see that you found the application of Right for the Right Reason (RRR) methods to time series data, as well as the introduction of dual loss terms for time and frequency domains, to be novel contributions. We would like to address your comments and clarify a few points regarding novelty, clarity, and the questions raised.
>
> **(W1) Novelty:** While we understand your observation that RioT leverages RRR methods applied to time series data, we respectfully highlight that our work introduces significant innovations in both methodology and application context.
> RioT extends RRR beyond vision-based applications by tailoring each of its components to time series data. In particular, adapting feedback from the frequency domain is essential in this context. Further, taking the step to include feedback from transformed representations overcomes one of the major limitations of RRR and previous XIL methods in general: Employing this technique enables feedback on confounding factors that are not separable in the original input domain. Previously, giving such feedback was not possible, but RioT enables this.
>
> **(Q1) Equation 1 $\bar{x}$**: The variable $\bar{x}$ is indeed the baseline for IG. We are sorry that this was not clear, and we added a clarification to the paper.
> In the experiments, we chose a zero baseline, following previous work [1, 2].
>
> **(Q2) Calculation of $e'_i(x)$:** The explanation $e'_i(x)$ is the model explanation for time step $i$ of the forecasting window. We follow the typical explanation computation for classification to obtain them but replace the target class with the forecasting output at time step $i$. Concretely, this corresponds to using IG by modifying Eq. 1 and replacing the target for the gradient computation with the forecasting output.
> > We made this more clear in the paper
>
>
> **(Q3) Requirement of the 2nd order derivative:**
> As RioT includes an optimization "through" the IG explanation, we need to compute the second-order derivative. In fact, we need the computation of the second-order mixed-partial derivative, where we first compute the explanation (first-order gradient w.r.t. the input) and then backpropagate through this w.r.t. the model parameters. In A.3 in the appendix, we discuss this derivative and its computational costs in more detail.
>
> **(Q4) Improving Figure 4 Visualization:** We apologize for any difficulty interpreting the color attributions in Figure 4. We enhanced the figure's contrast to improve visual clarity and updated it in the paper.
>
> **(Q5) Confounder Masks in Both Domains:** In the "Removing Multiple Confounders at Once" experiment, we indeed provide separate masks for spatial and frequency domains to capture different confounding factors across each domain. (Although time and frequency domains are related, unique confounding features may only appear prominently in one domain. Giving separate feedback in the respective domain helps the model learn to ignore spurious patterns in either domain.)
>
>
> [1] Dominique Mercier, Jwalin Bhatt, Andreas Dengel, and Sheraz Ahmed. Time to Focus: A Comprehensive Benchmark Using Time Series Attribution Methods. ArXiv:2202.03759, 2022.
>
> [2] Manjunatha Veerappa, Mathias Anneken, Nadia Burkart, and Marco F. Huber. Validation of XAI explanations for multivariate time series classification in the maritime domain. Journal of Computational Science, 58:101539, 2022.

---

> > ### Comment · Reviewer_EVUz · 2024-11-25
> > **Thank you for the rebuttal**
> >
> > Thank you for your response and for the clarifications. My concerns have been addressed.
> >
> > My score remains as is; I recommend acceptance.

---

> > > ### Author Response · Authors · 2024-11-27
> > >
> > > We are pleased to hear that we were able to address your remaining concerns. Considering that you found the paper to exhibit high originality, quality, and significance — particularly through the introduction of P2S and the confounded setup of UCR/UEA — and that you recommend its acceptance, we kindly ask you to consider raising your score.

---

### Author Response · Authors · 2024-11-16
**General Response**

We thank the reviewers for their time and the thoughtful reviews. We are especially happy to hear that the reviewers think that the approach of our framework that applies feedback on time series models via dual-domain explanations is novel, has high practical relevance, and is backed by a robust evaluation. We are content to hear that P2S is also considered a valuable contribution to the ML community.

In the following, we provide responses that are relevant to all reviewers. We then address each reviewer’s specific opportunities for improvement and questions individually. All revisions to the paper have been highlighted in blue for ease of reference.

---

> ### Author Response · Authors · 2024-11-16
> **Utilizing Human Feedback**
>
> Since multiple reviewers raised concerns regarding the reliance on human feedback, we would like to address this point comprehensively. Our work introduces a novel framework that enables the revision of time series models based on their explanations. Notably, the approach to obtaining feedback is flexible and can be tailored to the specific domain and resources available.
>
>
> **Effective with Minimal Feedback**
> Our experiments demonstrate that RioT performs effectively even when feedback is provided for only a small fraction of the data. As shown in Fig. 5, annotations on as little as 5% of the samples significantly mitigate confounders, yielding substantial improvement compared to scenarios with no feedback. This efficiency ensures that the feedback requirement is manageable.
>
> **Leveraging Generalizable Feedback to Reduce Effort**
> In many cases, annotations can be generalized across samples, further reducing the interaction cost. For example, in the P2S experiment, the domain expert provided feedback on only 5–10 samples, and this information was successfully extrapolated to all data points. Confounders, such as those with physical origins identified in P2S, often exhibit systematic patterns across data [1]. Once such patterns are identified in a subset of samples, they can be applied broadly without needing individual review. This systematic nature of confounders is a practical advantage that can be leveraged to minimize the overall annotation effort. Additionally, alternative strategies such as semi-automated feedback, transfer learning, or the use of large language models (LLMs) could further reduce the overhead associated with human annotations.
>
> **Domain-Specific Advantages of Expert Feedback**
> It is also important to consider the domain-specific costs associated with feedback. Reviewer concerns seem to assume that obtaining expert annotations is inherently expensive. However, in many real-world domains, expert feedback may actually be more accessible and cost-effective than other strategies to reduce confounding via data collection. For instance, privacy constraints and limited patient availability in the medical field make data collection challenging. Similarly, in mechanical engineering, experimental data acquisition can involve significant costs and complex setups. In such scenarios, leveraging domain expertise can be a practical and valuable alternative to extensive data collection efforts to mitigate confounding.
>
> **Strategies for Managing Unknown Confounders**
> We acknowledge that providing feedback on unknown confounders can be challenging. Nonetheless, existing research on confounder detection in machine learning [2, 3] could be adapted for time series data, facilitating the identification of such confounders. Even when a confounder is only partially identified, our experiments (cf. Fig. 4) show that RioT still achieves performance improvements. We agree that mitigating unknown confounders remains a broader challenge requiring further investigation and development in future work.
>
> **Advantages of Feedback in the Frequency Domain**
> While feedback in the frequency domain may initially appear non-intuitive, it can offer distinct advantages depending on the data type. For example, in audio data, the frequency representation often provides clearer insights, making it easier to annotate confounders. As demonstrated in [4], this perspective underscores that the frequency domain can sometimes be more intuitive and effective than the spatial domain, depending on the application context.
>
> In conclusion, we believe these points demonstrate that the reliance on human feedback is not a limitation of our framework but rather a flexible feature that can be adapted to different domains. Our method is designed to work effectively with minimal feedback, and the generalizability of annotations, along with domain-specific considerations, makes RioT a practical and scalable solution.
>
> [1] Lapuschkin, Sebastian, et al. "Unmasking Clever Hans predictors and assessing what machines really learn." Nature communications 10.1 (2019): 1096.
>
> [2] Yu Yang, Eric Gan, Gintare Karolina Dziugaite, and Baharan Mirzasoleiman. 2024. Identifying spurious biases early in training through the lens of simplicity bias. In International Conference on Artificial Intelligence and Statistics. PMLR,
> 2953–2961.
>
> [3] Shunxin Wang, Raymond Veldhuis, Christoph Brune, and Nicola Strisciuglio. 2023. What do neural networks learn in image classification? a frequency shortcut perspective. In Proceedings of the IEEE/CVF International Conference on Computer Vision. 1433–1442
>
> [4] Johanna Vielhaben, Sebastian Lapuschkin, Grégoire Montavon, and Wojciech Samek. Explainable AI for Time Series via Virtual Inspection Layers. ArXiv:2303.06365, 2023

---

> > ### Author Response · Authors · 2024-11-16
> > **Early Stopping as Baseline**
> >
> > As Reviewer 4ai3 pointed out, evaluating how well a simple early-stopping approach could mitigate the confounders is an interesting question. Intuitively, early stopping with an unconfounded validation set can avoid learning the confounder; however, as confounders are often learned early during training [5], this negatively impacts the performance of the target task. To validate this intuition, we performed an additional experiment where we evaluated the base model with early stopping based on an unconfounded validation set. We added the results to the appendix and also present them here. We can observe that early stopping (ES) can help in some instances to achieve performances similar to RioT (e.g., PatchTST with a frequency confounder or FCN with a spatial confounder). However, in most cases, the performance of early stopping is substantially lower than that of RioT, signaling that early stopping alone is not a sufficient approach to overcome confounders.
> >
> > [5] Yu Yang, Eric Gan, Gintare Karolina Dziugaite, and Baharan Mirzasoleiman. 2024. Identifying spurious biases early in training through the lens of simplicity bias. In International Conference on Artificial Intelligence and Statistics. PMLR, 2953–2961.

---

> ### Author Response · Authors · 2024-11-16
> **Early Stopping as Baseline**
>
> ## Early Stopping on forecasting datasets
>
> | Config                   | PatchTST, Train   | PatchTST, Test    | TiDE, Train       | TiDE, Test        |
> |:-------------------------|:------------------------|:------------------------|:------------------------|:------------------------|
> | Reference                | $0.26 \pm \tiny 0.01$ | $0.23 \pm \tiny 0.00$ | $0.27 \pm \tiny 0.01$ | $0.26 \pm \tiny 0.02$ |
> | SP Conf                  | $0.29 \pm \tiny 0.01$ | $0.96 \pm \tiny 0.03$ | $0.28 \pm \tiny 0.01$ | $1.19 \pm \tiny 0.03$ |
> | $\text{RioT}_\text{sp}$   | $0.44 \pm \tiny 0.00$ | $0.45 \pm \tiny 0.01$ | $0.53 \pm \tiny 0.02$ | $0.52 \pm \tiny 0.02$ |
> | $\text{ES}_\text{sp}$     | $0.48 \pm \tiny 0.05$ | $0.68 \pm \tiny 0.03$ | $1.20 \pm \tiny 0.25$ | $0.81 \pm \tiny 0.08$ |
> | Freq Conf                | $0.04 \pm \tiny 0.00$ | $0.53 \pm \tiny 0.05$ | $0.07 \pm \tiny 0.01$ | $0.34 \pm \tiny 0.08$ |
> | $\text{RioT}_\text{freq}$ | $0.71 \pm \tiny 0.10$ | $0.38 \pm \tiny 0.06$ | $0.07 \pm \tiny 0.01$ | $0.21 \pm \tiny 0.02$ |
> | $\text{ES}_\text{freq}$   | $0.48 \pm \tiny 0.09$ | $0.49 \pm \tiny 0.08$ | $0.21 \pm \tiny 0.08$ | $0.36 \pm \tiny 0.09$ |
>
>
> ## Early Stopping on classification datasets
>
> | Config                   | FCN, Train      | FCN, Test       | OFA, Train      | OFA, Test       |
> |:-------------------------|:----------------------|:----------------------|:----------------------|:----------------------|
> | Reference                | $0.99 \pm \tiny 0.00$ | $0.99 \pm \tiny 0.00$ | $1.00 \pm \tiny 0.00$ | $0.98 \pm \tiny 0.02$ |
> | SP Conf                  | $1.00 \pm \tiny 0.00$ | $0.74 \pm \tiny 0.06$ | $1.00 \pm \tiny 0.00$ | $0.53 \pm \tiny 0.02$ |
> | $\text{RioT}_\text{sp}$   | $0.98 \pm \tiny 0.01$ | $0.93 \pm \tiny 0.03$ | $0.96 \pm \tiny 0.08$ | $0.98 \pm \tiny 0.01$ |
> | $ES_\text{sp}$     | $0.87 \pm \tiny 0.01$ | $0.91 \pm \tiny 0.03$ | $0.69 \pm \tiny 0.02$ | $0.67 \pm \tiny 0.04$ |
> | Freq Conf                | $0.98 \pm \tiny 0.01$ | $0.87 \pm \tiny 0.03$ | $1.00 \pm \tiny 0.00$ | $0.72 \pm \tiny 0.02$ |
> | $\text{RioT}_\text{freq}$ | $0.94 \pm \tiny 0.00$ | $0.90 \pm \tiny 0.03$ | $0.96 \pm \tiny 0.02$ | $0.98 \pm \tiny 0.02$ |
> | $ES_\text{freq}$   | $0.83 \pm \tiny 0.01$ | $0.86 \pm \tiny 0.02$ | $0.81 \pm \tiny 0.01$ | $0.75 \pm \tiny 0.02$ |

---

### Author Response · Authors · 2024-11-22
**Discussion Period is Ending Soon**

Dear Reviewers,

in line with all changes made to the paper, we hope that we have resolved all questions and concerns. If there are any further comments or suggestions, we are happy to address them before the rebuttal period ends.

Best regards,

The Authors

---

### Meta-Review · Area_Chair_xjFF · 2024-12-19

**Metareview:**

The paper presents RioT to mitigate confounders in time series models. Strengths include novel application of RRR to time series, new dataset P2S, and empirical evidence. Weaknesses are RioT's lack of novelty in concept, dependency on human feedback, training cost, and some clarity and generalization issues. Overall, it has value but needs improvement in addressing concerns about human feedback, experimental design, and writing, which suggests a reject.

**Additional Comments On Reviewer Discussion:**

Reviewers noted strengths like innovation and empirical results but also raised concerns such as the need for human feedback, training cost, and clarity of explanations. Authors responded, providing clarifications and additional experiments. Some concerns from, such as acPg
 and 4ai3, were not fully resolved. I

---

### Decision · Program_Chairs · 2025-01-22

Reject